# All electromagnetic scattering bodies are matrix-valued oscillators

Lang Zhang[1], Francesco Monticone [2] & Owen D. Miller [1] ✉

Scattering theory is the basis of all linear optical and photonic devices, whose spectral response underpins wide-ranging applications from sensing to energy conversion. Unlike the Shannon theory for communication channels, or the Fano theory for electric circuits, understanding the limits of spectral wave scattering remains a notoriously challenging open problem. We introduce a mathematical scattering representation that inherently embeds fundamental principles of causality and passivity into its elemental degrees of freedom. We use this representation to reveal strong constraints in the mathematical structure of scattered fields, and to develop a general theory of the maximum radiative heat transfer in the near field, resolving a long-standing question. Our approach can be seamlessly applied to high-interest applications across nanophotonics, and appears extensible to general classical and quantum scattering theory.

Probing and harnessing the frequency dependence of electromagnetic scattering underlies atomic spectroscopy, molecular sensing, information and energy technologies, and more[1–4]. A key pillar of electromagnetic scattering theory is the decomposition of scatterers into "resonators," in which spectral response is determined by lifetimes and coupling coefficients (or suitable generalizations) of resonant modes[5–8]. These "physical oscillators" enable complex scenarios to often be well-described by a small number of parameters, and they offer high-accuracy descriptive modeling. However, there is typically no limit on the possible number, lifetimes, or couplings of the modes, such that little can be said about their extreme limits. Mathematically, the difficulty in finding extreme limits arises because the set of all possible resonator designs is nonconvex. Hence physical oscillators provide little prescriptive guidance: what lineshapes are physically possible, and what are the ultimate limits of corralling broadband radiation?

In lieu of resonator decompositions, passivity, and causality have long been recognized as key constraints on broadband response in linear physical systems without gain[9–15]. Causality is implied by passivity, so that one need not separately invoke it, and the foundations of linear system theory typically start with passivity[12]. Passivity-based approaches to spectral response have yielded fundamental limits for matching networks in circuit theory[16,17], optical attenuation (e.g., in

stellar grains[18]), material susceptibilities[19,20], and more[15]. Yet passivity itself is not a panacea, and electromagnetic scattering theory is a domain where its application has been met with limited success. Special linear-amplitude, "optical-theorem"-like power quantities have bounds analogous to those for optical attenuation[21,22]. But the general scattering properties of arbitrary systems are described by scattering matrices $\mathbb{S}$ that map input excitations at any number of "port" (power-carrying "channels" external to the scatterer) to their corresponding outputs, and scattering $\mathbb{S}$ matrices have few (if any) practical spectral limitations. Their analytic properties and representation theorems have been extensively studied, from dispersion relations[13] and Blaschke-product representations[13] to existence theorems for poles, zeros, and their generalizations[23], but known representations suffer from the same issue as their coupled-mode counterparts: their degrees of freedom reside in nonconvex (and often unbounded) sets. This makes it difficult or impossible to identify optimal response, or upper limits thereof, across the physical design spaces of scientists and engineers.

The potential value of spectral-response bounds is highlighted by a long-standing question in energy transport: what is the maximum rate at which two bodies can radiatively exchange heat in the near field? Going back many decades, it has been understood that radiative heat exchange in the near field can be substantially larger than its far-

[1]Department of Applied Physics and Energy Sciences Institute, Yale University, New Haven, CT 06511, USA. [2]School of Electrical and Computer Engineering, Cornell University, Ithaca, NY 14853, USA. ✉e-mail: owen.miller@yale.edu

field counterpart[24–26], due to the enormous number of accessible evanescent channels in addition to propagating ones, yet the maximum extent of this enhancement−with ramifications for applications such as thermophotovoltaics[27,28], photonic refrigeration[29], and heat-assisted magnetic recording[30]−has been far less clear. Previous theoretical bounds[22,31–34] have suggested strong material-electron-density dependencies, unbounded response for low-loss materials, and orders-of-magnitude gaps from known designs (>750X). The computational complexity of the problem has prohibited the application of large-scale inverse design techniques, leaving unresolved whether current designs are sub-optimal or the bounds are too loose.

In this article, we show that an alternative scattering matrix, the $\mathbb{T}$ matrix[35], can be represented by fictitious "mathematical oscillators" that are ideally suited for probing optimal spectral response. We show that passivity, in tandem with the specific interaction characteristics of materials with electromagnetic waves in low- and high-frequency limits, leads to $\mathbb{T}$-matrix representations in terms of lossless Drude−Lorentz and Drude−Lorentz-like oscillators with matrix-valued (spatially nonlocal) coefficients. Crucially, the only degrees of freedom of these oscillators are their matrix-valued coefficients, which are constrained to a bounded, convex set. Such limitations must imply strong constraints on scattering response, which we use to identify a simple, general theoretical limit to near-field radiative heat transfer. Our approach offers insights into why planar structures are better than sharp-tip patterns, why unconventional plasmonic materials should offer the largest enhancements, and yields material-independent bounds within a small factor (5X) of state-of-the-art designs.

## Results

### Passivity constraints and oscillator representation

In linear, time-invariant electrodynamics, the $\mathbb{T}$ matrix is the linear operator that relates electromagnetic fields incident upon a scatterer to the polarization fields they induce[35]. For simplicity of notation and exposition, we assume any standard spatial numerical discretization of sufficiently high accuracy; we collate the incident fields $\mathbf{E}_{inc}(\mathbf{x})$ into a vector $\mathbf{e}_{inc}$ and the polarization fields $\mathbf{P}(\mathbf{x})$ into a vector $\mathbf{p}$, so that the frequency-domain ($e^{-i\omega t}$ time convention) $\mathbb{T}$-matrix is defined by

$$\mathbf{p}(\omega) = \mathbb{T}(\omega)\mathbf{e}_{inc}(\omega), \tag{1}$$

the discrete analog of the convolution equation $\mathbf{P}(\mathbf{x}, \omega) = \int \mathbb{T}(\mathbf{x}, \mathbf{x}', \omega)\mathbf{E}_{inc}(\mathbf{x}', \omega)\,d\mathbf{x}'$. The $\mathbb{T}$ matrix can be derived from first principles via integral operators (cf. Supplementary Note 1 or ref. 35), and its time derivative (or product with $-i\omega$) can be interpreted as an admittance matrix.

A passive scatterer in vacuum has a causal response function, such that it is analytic in the upper-half plane and satisfied Kramers−Kronig (KK) relations[15]. We write the KK relation in terms of $\omega\mathbb{T}(\omega)$ to account for possible simple poles at zero: by Cauchy's residue theorem, for $\omega$ in the UHP, $\omega\mathbb{T}(\omega) = \frac{1}{i\pi}\int_{-\infty}^{\infty}\frac{\omega'\mathbb{T}(\omega')}{\omega'-\omega}\,d\omega'$. (Physically, $\mathbb{T}(\omega)$ must decay as $1/\omega^2$ at high frequencies, such that $\omega\mathbb{T}(\omega)$ is square integrable.) Taking the Hermitian part of this equation yields $\mathrm{Re}\,[\omega\mathbb{T}(\omega)] = \frac{1}{\pi}\int_{-\infty}^{\infty}\frac{\omega'\mathrm{Im}\,\mathbb{T}(\omega')}{\omega'-\omega}\,d\omega'$. Hence we can isolate the anti-Hermitian part of $\mathbb{T}(\omega)$ as its only degrees of freedom:

$$\begin{aligned}\omega\mathbb{T}(\omega) &= \mathrm{Re}\,[\omega\mathbb{T}(\omega)] + i\mathrm{Im}\,[\omega\mathbb{T}(\omega)]\\ &= \frac{1}{\pi}\int_{-\infty}^{\infty}\left[\frac{1}{\omega_i-\omega}+i\pi\delta(\omega-\omega_i)\right]\omega_i\mathrm{Im}\,\mathbb{T}(\omega_i)\,d\omega_i\\ &= \frac{1}{\pi}\lim_{\gamma\to 0}\int_{-\infty}^{\infty}\frac{1}{\omega_i-\omega-i\gamma}\omega_i\mathrm{Im}\,\mathbb{T}(\omega_i)\,d\omega_i.\end{aligned} \tag{2}$$

To further compress to positive frequencies only, we exploit symmetries of $\mathbb{T}(\omega)$. The Hermitian matrix $\mathbb{Z} = \omega_i\mathrm{Im}\,\mathbb{T}(\omega_i)$ can be separated into its reciprocal part $\mathbb{X} = (\mathbb{Z} + \mathbb{Z}^T)/2$ and its nonreciprocal part $\mathbb{Y} = (\mathbb{Z} - \mathbb{Z}^T)/2$. Real-valued time-domain fields require that

$\mathbb{T}(-\omega) = \mathbb{T}^*(\omega)$, which implies that $\mathbb{X}(-\omega) = \mathbb{X}(\omega)$ and $\mathbb{Y}(-\omega) = -\mathbb{Y}(\omega)$. Then algebraic manipulations of Eq. (2) give

$$\mathbb{T}(\omega) = \frac{2}{\pi}\lim_{\gamma\to 0}\int_0^{\infty}\frac{1}{\omega_i^2-\omega^2-i\gamma\omega}\left[\mathbb{X}(\omega_i) + \frac{\omega_i}{\omega}\mathbb{Y}(\omega_i)\right]d\omega_i. \tag{3}$$

We provide an alternative derivation of the same expression in the Methods section, by recognizing that $-i\omega\mathbb{T}$ is a passive admittance matrix, which implies a Herglotz−Nevanlinna representation[36] that can be reduced to Eq. (3). For any scattering problem there are at least six matrices that satisfy an expression similar to Eq. (3): a scattering matrix, an impedance matrix, and an admittance matrix, each defined either in the volume or on a bounding surface. Yet only one of those six −the volume admittance matrix (essentially, $\mathbb{T}(\omega)$)−appears to be useful for wave-scattering bounds. While Eq. (3) reduces the degrees of freedom to the anti-Hermitian part of $\mathbb{T}$, additional passivity considerations are needed to meaningfully constrain the possible scattering response.

The next constraints come directly from passivity. Passivity means that polarization fields do no net work. The work done by the incident fields on the polarization currents $\mathbf{J}$ is $\frac{1}{2}\mathrm{Re}\int\mathbf{E}_{inc}^*\cdot\mathbf{J} = \frac{1}{2}\mathrm{Im}\int\mathbf{E}_{inc}^*\cdot\omega\mathbf{P}$. Positivity of this expression implies that the anti-Hermitian part of $\omega\mathbb{T}(\omega)$ is positive semidefinite, which we write $\omega\,\mathrm{Im}\,\mathbb{T}(\omega) \geq 0$. (This is equivalent to the condition that admittance matrices have a positive semidefinite Hermitian part[15].) This means that $\mathbb{X}(\omega) + \mathbb{Y}(\omega) \geq 0$ for any real-valued $\omega$. Using the symmetry relations for $\mathbb{X}(\omega)$ and $\mathbb{Y}(\omega)$ around $\omega = 0$, we have the constraints $\mathbb{X}(\omega) + \mathbb{Y}(\omega) \geq 0$ and $\mathbb{X}(\omega) - \mathbb{Y}(\omega) \geq 0$ at positive frequencies, which further imply $\mathbb{X}(\omega) \geq 0$. These constraints are convex (though still unbounded) in $\mathbb{X}(\omega)$ and $\mathbb{Y}(\omega)$.

The final key element is the identification of sum rules. Sum rules typically come from evaluation of KK relations in the limit $\omega\to\infty$ or $\omega = 0$. At infinite frequency, the electrons of a material can be regarded as free, and material susceptibilities must scale as $\chi(\omega) \to -\omega_p^2/\omega^2$, where $\omega_p^2$ is proportional to the total electron density of the material[13]. In this limit, the first Born approximation is asymptotically exact, and the polarization field is given by $\mathbf{P} \simeq \chi\mathbf{E}_{inc} \simeq -(\omega_p^2/\omega^2)\mathbf{E}_{inc}$ (in units where the free-space permittivity is 1), implying that the $\mathbb{T}$ matrix asymptotically approaches $-(\omega_p^2/\omega^2)\mathbb{I}_V$, where $\mathbb{I}_V$ is the identity matrix on the scatterer volume $V$. Inserting this limit into the KK relation derived before Eq. (2) yields the high-frequency sum rule,

$$\int_{-\infty}^{\infty}\omega\,\mathrm{Im}\,\mathbb{T}(\omega)\,d\omega = 2\int_0^{\infty}\mathbb{X}(\omega)\,d\omega = \pi\omega_p^2\mathbb{I}_V. \tag{4}$$

This sum rule constrains the total contributions from $\mathrm{Im}\,\mathbb{T}(\omega)$ over all frequencies, a spatially resolved scattering generalization of the $f$ sum rule for material-susceptibility oscillator strengths[37–39]. The nonreciprocal $\mathbb{Y}$ matrix makes no contribution to the integral due to its odd symmetry around $\omega = 0$. Similarly, the low-frequency asymptote is known: we can write $\mathbb{T}(\omega \to 0) = \mathbb{T}_{0,V}$, where $\mathbb{T}_{0,V}$ is a Hermitian positive semidefinite matrix in the static limit. Inserting this expression into the KK relation derived before Eq. (2) yields a low-frequency sum rule,

$$\int_{-\infty}^{\infty}\frac{\mathrm{Im}\,\mathbb{T}(\omega)}{\omega}\,d\omega = 2\int_0^{\infty}\frac{\mathbb{X}(\omega)}{\omega^2}\,d\omega = \pi\mathbb{T}_{0,V}. \tag{5}$$

For design problems, one considers many possible scatterer domains $V$, each of which has different matrices on the right-hand sides of the sum rules of (Eqs. (4), (5). How, then, can one accommodate many possible designs? Here we can again make the (critical) choice of the Hermitian/anti-Hermitian split in the KK relation, which, as we prove in Methods, endows the sum rules with a monotonicity property: enlarging $V$ can only increase (in a positive semidefinite sense) the right-hand sides of Eqs. (4), (5). Hence, for a designable

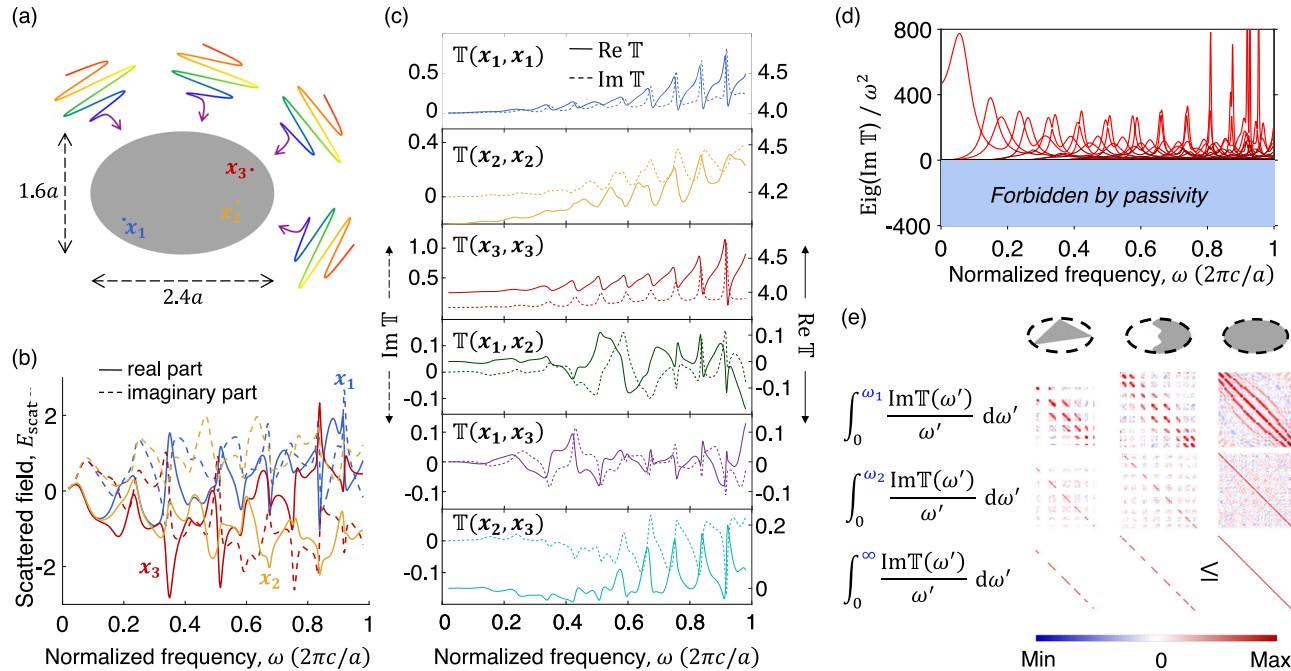

**Fig. 1 | Oscillator structure in broadband scattering. a** Schematic depiction of plane waves incident upon an elliptical scatterer. **b** Scattered fields at points 1–3 of (**a**), exhibiting seemingly random variations for a single plane wave incidence. **c** Real and imaginary parts of the $\mathbb{T}(\omega)$ matrix elements corresponding to the points in (**a**), showing oscillator-like responses consistent with the KK relation. **d** All eigenvalues of $\mathbb{T}(\omega)$ are positive at all frequencies, as a consequence of passivity.

**e** Convergence of a sum-rule integral for three upper limits ($\omega_1 = 0.2$, $\omega_2 = 0.4$, units of $2\pi c/a$), for three scatterers within an elliptical designable domain. The sum rules converge to a low-frequency matrix constant (in this TE-polarization case, a scalar multiple of the identity), and the constants for the two smaller scatterers obey the domain monotonicity property. The combined elements of (**c–e**) impose strong constraints on broadband scattering.

domain $D$ containing all possible scatterer sub-domains, we can convert the equalities of Eqs. (4), (5) for specific volumes $V$ into inequalities over the designable domain $D$.

We can unify the above properties to create a framework for fundamental limits. The $\mathbb{T}(\omega)$ matrix can always be written in the form of Eq. (3), while the real-symmetric matrix $\mathbb{X}(\omega)$ and the skew-symmetric matrix $\mathbb{Y}(\omega)$ are strongly constrained. We renormalize $\mathbb{X} \rightarrow (\pi/2)\mathbb{X}$ and $\mathbb{Y} \rightarrow (\pi/2)\mathbb{Y}$ to simplify the oscillator representation. Together, for the $\mathbb{T}$ matrix of any designed scatterer within a designable region $D$, we have:

$$\mathbb{T}(\omega) = \lim_{\gamma \to 0} \int_0^\infty \frac{1}{\omega_i^2 - \omega^2 - i\gamma\omega} \left[ \mathbb{X}(\omega_i) + \frac{\omega_i}{\omega} \mathbb{Y}(\omega_i) \right] d\omega_i,$$
$$\mathbb{X}(\omega_i) \geq 0, \quad -\mathbb{X}(\omega_i) \leq \mathbb{Y}(\omega_i) \leq \mathbb{X}(\omega_i), \quad (6)$$
$$\frac{1}{\omega_p^2} \int_0^\infty \mathbb{X}(\omega_i) \, d\omega_i \leq \mathbb{I}_D, \quad \int_0^\infty \frac{1}{\omega_i^2} \mathbb{X}(\omega_i) \, d\omega_i \leq \mathbb{T}_{0,D}.$$

The collective representation of Eq. (6) is the foundational result of our paper: the $\mathbb{T}$ matrix of any linear scattering body must be decomposable into a set of lossless oscillators, with matrix-valued coefficients satisfying definiteness conditions and constrained in total strength. The only degrees of freedom in the scattering process are the matrices $\mathbb{X}(\omega_i)$ and $\mathbb{Y}(\omega_i)$, both of which have strong constraints on the bandwidth over which they can be nonzero. The $\mathbb{T}(\omega)$ matrix is linear in these matrix degrees of freedom and the constraints are bounded convex sets. Hence this representation encodes the constraints of passivity and sum rules for electromagnetic scatterers in a mathematical structure that is ideally suited for optimization and fundamental limits.

For a first demonstration of the mathematical structure implied by this representation, we consider broadband scattering from an elliptical dielectric cylinder. To clarify the origin of the oscillators, we use a material with $\chi = \omega_p^2/(\omega_0^2 - \omega^2 - ig\omega)$, with $\omega_p = 20$, $\omega_0 = 10$, $g = 0.01\omega_p$,

which is nearly dispersionless with $\chi = 4$ for $\omega$ between 0 and 1 (all frequencies in unit of $2\pi c/a$) and consistent with the necessary high-frequency asymptotic response. The scattered electric field at various points within the scatterer, computed by full-wave simulations (cf. Supplementary Note 6), is shown in Fig. 1b, but is hard to interpret due to its seemingly random undulations. Advances in quasinormal-mode (QNM) techniques suggest that one could accurately reproduce these fields with a modest number of QNMs[8], but that modeling capability does not imply an understanding of the extreme limits of what is possible. How many resonances can be excited? With what amplitudes, phases, and overlaps with power-carrying channels?

By contrast, consider the lineshapes of the Hermitian and anti-Hermitian parts of the $\mathbb{T}$ matrix (computed on a discretization of more than 37,000 spatial degrees of freedom), as depicted in Fig. 1c for the same three spatial locations and their cross terms. The lineshapes of the $\mathbb{T}$-matrix elements closely mimic the Drude-Lorentz-like behavior of electronic transitions, but they arise not from real material oscillators, but from complex wave-scattering behavior itself. The first three traces of Fig. 1c clearly show positive imaginary parts of varying widths, and real parts that transition from minima to maxima between the peaks of the imaginary parts, then transitioning back to minima where the imaginary parts peak. Hence the peaks tend to coincide (with the real-part peak slightly preceding the imaginary-part peak), and the characteristic lineshapes might be described as minima-to-maxima-to-transition for the real parts and Lorentzian-like for the imaginary parts. The second set of three traces in Fig. 1c do not have exactly this pattern, because they have complex-valued residues that mix the real and imaginary parts. But their underlying "oscillator-like" structure is still visible: one still sees peaks in one part nearly coinciding with (but slightly preceding) peaks in the counterpart, as well as Lorentzian-like lineshapes in one part being paired with minima-to-maxima-to-transition lineshapes in the other. By contrast, no such structure arises in the scattered fields

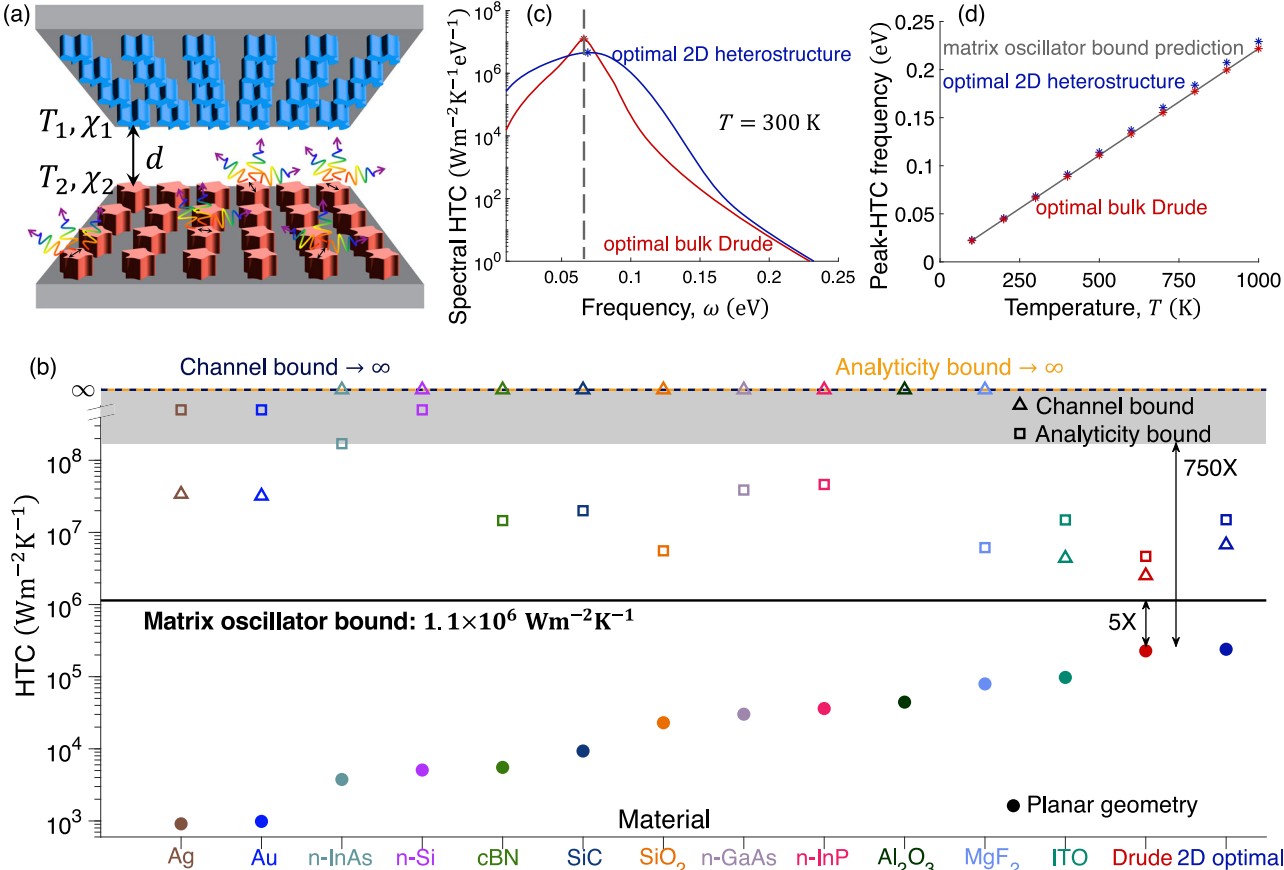

**Fig. 2 | Fundamental limits to NFRHT. a** NFRHT between two closely separated bodies. **b** Heat-transfer coefficients of planar bodies comprising high-performance planar-body geometries (filled circles), in comparison with the best previous theoretical bounds[22,34] (open squares and triangles). The previous bounds diverged for some materials, while showing enormous gaps (>750X) for others, and their trendline seems to decrease left-to-right, whereas planar-body performance increases. In black is the theoretical bound offered by our $\mathbb{T}$-matrix representation, very close to the best possible planar bodies. **c** Spectral HTC of the best bulk (Drude, red) and 2D (heterostructure, blue) designs, with spectral peaks marked by asterisks and the predicted theoretical peak from our oscillator-based bound (gray dashed line). **d** Peak-HTC frequency of optimal materials (red, blue), nearly coinciding with the predictions of the bound (gray) for a wide range of temperatures.

of Fig. 1b, because they simply do not have a representation resembling Eq. (6).

Collectively, the lineshape widths of the $\mathbb{T}$-matrix elements are nonzero thanks to the underlying resonant physics, but every frequency can and should (for our purposes) be interpreted as having its own, lossless-oscillator amplitude, given by $\omega \operatorname{Im} \mathbb{T}(\omega)$. The diagonal components have imaginary parts that must be positive. The off-diagonal components need not have positive imaginary parts, but they are constrained by the positive-definiteness requirements of the entire matrix, as verified in Fig. 1d, which shows the positivity of the eigenvalues of $\operatorname{Im} \mathbb{T}(\omega)$. The final key component for meaningful constraints from such a representation is the sum rules, and their domain monotonicity property. Figure 1e shows the integrated response for three scattering bodies within the elliptical designable domain, showing both their convergence to the appropriate sum-rule matrix constant as the integral is taken to infinity (the numerical integral converges to <1.7% error, as measured by the matrix Frobenius norm, using a 2000-point Gauss-Legendre quadrature for frequencies from 0 to $40(2\pi c/a)$), as well as the satisfaction of domain monotonicity between the sum-rule matrices for the two sub-domains of the elliptical domain. As a whole, these combined elements offer an ideal representation for identifying fundamental limits to spectral control.

## Ultimate limits to NFRHT
Next, we apply our formulation to the question of maximal NFRHT. NFRHT, as depicted in Fig. 2a, poses prohibitive computational challenges—spatially and temporally incoherent, broadband thermal sources, exciting rapidly decaying near fields over large macroscopic areas—which have limited previous design efforts primarily to high-symmetry structures such as planar bodies[40–42]. Numerous approaches have identified particular constraints with corresponding theoretical bounds[22,31–34], but as we show in Fig. 2b, there are orders-of-magnitude differences between the best structures and the best bounds[22,34]. We label the bounds by their distinguishing attributes: in ref. 22 ("analyticity bound"), complex-analyticity played a central role, while in ref. 34 ("channel bound"), a decomposition into power-carrying channels was the starting point. Recently, it was discovered that a set of unconventional plasmonic materials offer significant (10X) improvements over the previous best planar structures[43], but otherwise, the field has been at an impasse, without a meaningful approach to either improve the best designs or tighten the bounds.

The $\mathbb{T}$ matrix formulation resolves this impasse. The heat transfer coefficient (HTC) between two bodies is the net flux rate (per area and per degree $K$) of electromagnetic energy passing between bodies at temperatures $T$ and $T + \Delta T$, as measured by the integral of power flux $(1/2)\operatorname{Re}(\mathbf{E} \times \mathbf{H}^* \cdot \hat{\mathbf{n}})$ through a separating plane with normal vector $\hat{\mathbf{n}}$. The incoherent sources in body $i$ with temperature $T_i$ and susceptibility $\chi_i(\omega)$, by the fluctuation-dissipation theorem[40], are given by $\langle J_j(\mathbf{x},\omega) J_k^*(\mathbf{x}',\omega) \rangle = (4\varepsilon_0 \omega/\pi)\Theta(\omega,T_i)\operatorname{Im}[\chi_i(\omega)]\delta_{jk}\delta(\mathbf{x}-\mathbf{x}')$ at frequency $\omega$, where $\Theta(\omega,T_i) = \hbar\omega/(e^{\hbar\omega/k_B T_i} - 1)$ is the Planck spectrum, and $k_B$ is the Boltzmann constant. There are a variety of mathematical transformations that we make to this problem to make it more amenable to

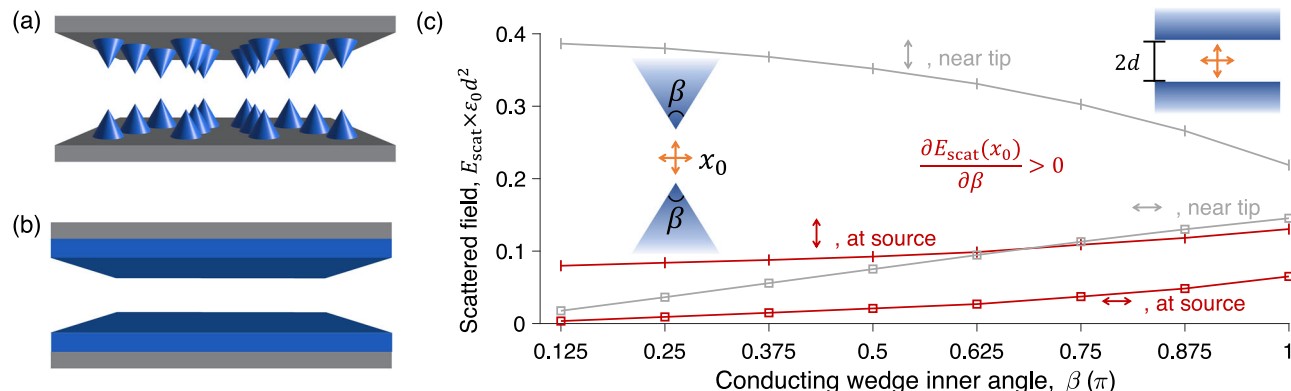

**Fig. 3 | Sharp tips do not enhance broadband NFRHT.** For NFRHT, an important question is whether the optimal structure might be (**a**) patterned with sharp cones, or (**b**) planar structures without any patterning. **c** Numerical simulations of the scattered fields from an electrostatic dipole source between two-dimensional conducting wedges, either at the source location (red, distance $d$ from either tip) or near the tip (gray, distance $0.1d$). Near-tip scattered fields (gray) exhibit the lightning-rod effect, increasing with tip sharpness. But frequency-integrated energy transfer is proportional to the scattered field at the source (red), which increases with wedge angle, consistent with domain monotonicity. Out of all possible sub-domains, planar half-spaces must have the largest frequency-integrated response.

optimization, detailed in Methods, such as using reciprocity to move the sources out of the hotter body and onto the dividing surface, exploiting spatial symmetries of the bounding domains (two half-spaces, allowing for any patterning within), as well as a near-field generalization of the "optical theorem"[44]. The key novelty, however, is our use of Eq. (6): once we have transformed the problem to an appropriate function of the two-body $\mathbb{T}$ matrix, we insert the representation theorem of $\mathbb{T}$ as a sum of positive-semidefinite matrix coefficients with Drude–Lorentz lineshapes. NFRHT at moderate or even high temperatures is dominated by low-frequency response, so we only impose the low-frequency sum rule. For a designable domain $D$ of two-half spaces, $\mathbb{T}_{0,D} = \alpha \mathbb{I}_D$, where $\alpha$ is a scalar function of the material susceptibility that is bounded above by 2. Once we insert the $\mathbb{T}$-matrix representation into the NFRHT expression, the resulting optimization problem over the infinite set of matrix oscillator coefficients has an analytical upper bound. Straightforward algebraic manipulations (cf. Methods) lead to an ultimate limit to near-field radiative HTC given by

$$\text{HTC} \le \beta \frac{T}{d^2}, \qquad (7)$$

where $d$ is the minimum separation between the bodies, $T$ the temperature of the cooler body, and $\beta = 0.11(\alpha k_B^2/\hbar) = 3.8 \times 10^5 \, \text{Wnm}^2/\text{m}^2/\text{K}^2$, a numerical constant. This limit cannot be surpassed by any geometric patterning, nor can exotic optical properties of any material alter its value.

Figure 2b compares our theoretical limit with the current state-of-the-art, as well as the best known bounds. Whereas the gap between the optimal planar structures and the best previous bounds was at least 750X (and diverging to ∞ for some materials), the expression of Eq. (7) is only 5X larger than the best design. This bound has no material dependence, which resolves the problematic trend that if one orders the materials by their planar performance, as in Fig. 2b, the previous bounds tended to predict worse maximal performance from left to right. The resolution of this discrepancy is our use of the low-frequency sum rule, which encodes a constraint on the local density of states seen by thermal emitters that depends only on their gap separation, independent of material. The $\mathbb{T}$-matrix approach predicts an optimal NFRHT frequency of $\omega_{max} = 2.57 \frac{k_B T}{\hbar}$, determined by the overlap of the Planck function with the Drude–Lorentz lineshape. The predictions are matched almost exactly by computationally optimized planar Drude metals or 2D heterostructures, as shown in Fig. 2c, d. For 300 K temperature, the spectra shown in Fig. 2c peak at almost exactly

the optimal oscillator frequency, and the match persists across all relevant temperatures, as shown in Fig. 2d.

Although it seemed plausible (even likely) that nano-structuring may lead to enhanced NFRHT through field-concentration (lightning-rod) effects, our sum rule explains why this is not the case: sharp tips can enhance the fields very close to a sharp tip, but not at the source location itself. The local density of states is proportional to the latter, and hence is not enhanced by lightning-rod effects. To illustrate why sharp-tip-based (or related) structures are inferior, we design a numerical experiment. In NFRHT, after using reciprocity, the incident field arises from point sources along a separating plane between the bodies. For a single given dipole, the relevant low-frequency sum rule constant is $\mathbf{e}_{\text{inc}}^T \mathbb{T}_0 \mathbf{e}_{\text{inc}} = \mathbf{e}_{\text{inc}}^T \mathbf{p}$, i.e., the overlap between the (static) incident field and the (static) induced polarization. This is equivalent to the scattered field at the point source.

Figure 3a, b compares schematic depictions of sharp-tip versus planar-area structures, while Fig. 3c shows finite-element calculations for two-dimensional analogs, with dipole sources of both possible polarizations between conducting wedges of arbitrary inner angles $\beta$, with the sources a distance $d$ from either tip. The gray lines show the scattered fields near the tip (at distances $0.1d$), which for the transverse polarization increase at smaller angles, i.e., sharper tips. This is the typical lightning-rod effect. Yet these amplified fields play no role in determining the total level of broadband energy transfer; the static constant controlling the sum rule is proportional to the scattered field back at the source, shown in red. This quantity increases with the wedge inner angle, a result that must be true by our domain monotonicity theorem. Hence planar bodies ($\beta = \pi$) must have the largest possible frequency-integrated response. The only remaining question is whether the frequency response can be tailored for maximum overlap with the Planck spectrum, but that question was answered above, affirmatively, by optimal material dispersion relations.

The closeness of the arbitrary-structure bound of Eq. (7) to the best planar structures arises despite quite different mathematical routes to these results. The translational symmetry of planar bodies implies conserved wavevectors and thus a set of evanescent plane-wave channels that are independent, with Landauer-like transmissivities[42]. Such an approach cannot describe patterned structures. Instead, Eq. (7) culminates after using (generalized) reciprocity to move the sources from the hot body to the dividing surface, the sum rule to encapsulate the maximum densities of states seen by those sources, and the $\mathbb{T}$-matrix representation to constrain the possible scattering lineshapes. The striking similarity of the two results suggests that even when confronted by spatial and temporal incoherence,

rapidly decaying fields, and large areas, the oscillator representation compactly captures the key physics of maximal response in the near field.

## Discussion

In this article, we have introduced a framework for broadband electromagnetic scattering. The example of Fig. 1 showcases the joint consequences of passivity and sum rules on the structure of the electromagnetic $\mathbb{T}(\omega)$ matrix. We propose a recipe for identifying fundamental limits: rewrite any objective of interest in terms of the $\mathbb{T}(\omega)$ matrix, and then use the representation of Eq. (6) as the constraints. Our application of this framework to NFRHT offers clear guidance for the fundamental limits of radiative heat transfer and the physical mechanisms underlying them. The generality of our $\mathbb{T}$ matrix representation offers tantalizing prospects for wide-ranging applications across nanophotonics. Metasurfaces[45,46], for example, offer a compact form factor for optics. A central question is the extent to which metasurfaces can control incoming waves[47–49], across varying frequency and angular bandwidths, for applications from lenses to virtual and augmented reality. Similarly, techniques for imaging through opaque media have flourished with modern spatial light modulators[50], with a key open question being ultimate limits to spectral control. In photovoltaics and photodetection, the quest for ever-thinner devices must ultimately contend with fundamental limits, and similarly across almost every application of nanophotonics. There has long been a need to quantify ultimate limits to spectral control; our approach offers a theory to do so.

Our approach also dovetails seamlessly with a recent flurry of activity in understanding the limits controlling spatial degrees of freedom in nanophotonic systems[51–56]. Transforming the typical Maxwell differential equations into a set of local conservation laws in space, for real and reactive power flows, leads to a mathematical form of the design problem that is amenable to systematic approaches to computational bounds. For a single frequency (or a small number of them[57]), such conservation laws have shown powerful capabilities for identifying fundamental limits to spatial control. In these approaches, the degrees of freedom of the system are typically encoded not in the electric and magnetic fields, but rather in the electric and magnetic polarization currents that they induce. Those polarization fields are exactly those that are determined by the $\mathbb{T}$ matrix, which means that our spectral expansion of the $\mathbb{T}$ matrix should be seamlessly compatible with the spatial conservation laws proposed in ref. 52,53. Together, the two approaches may enable a complete understanding of the spatio-spectral limits of electromagnetic systems.

One might wonder why we have utilized the $\mathbb{T}(\omega)$ matrix, when the vast majority of photonics theory uses the scattering matrix $\mathbb{S}(\omega)$? There are two reasons. First, in many scattering systems, incoming and outgoing waves are spatially distributed (e.g., spherical waves), requiring exquisite care with $\mathbb{S}$-matrix causality conditions, leading to (for example) phase shifts in the KK relations[13]. It becomes unclear which degrees of freedom (if any) are necessary, sufficient, and have convex passivity constraints. The second issue is that there is not, as far as we know, a useful $\mathbb{S}$-matrix sum rule of a positive semidefinite quantity. Without such a sum rule, all response is unbounded. As discussed above, scatterer-volume $\mathbb{T}$ matrices appear to be the unique scattering/impedance/admittance matrix where KK relations, passivity, and sum rules can all be combined into a bounded, convex set of constraints.

More broadly, the insight at the foundation of our framework, about the mathematical properties of scattering $\mathbb{T}$ matrices, can be directly applied to any classical wave equation. These techniques should be readily extensible to linear scattering problems in acoustics, elasticity, fluid dynamics, and beyond. The mathematical structure of the wave equation is similar in each case, and the resulting $\mathbb{T}$ matrices should therefore have similar representations. An interesting twist may arise in acoustic scattering theory, where materials with higher-than-vacuum speeds of sound lead to "non-causal" scattering processes[58] that have prevented the development of classical sum rules, and would appear to prohibit a corresponding $\mathbb{T}$ matrix representation. Yet the $\mathbb{T}$ matrix itself may offer a new route to complex-analytic response functions in exactly such scenarios. The reason higher sound speeds lead to "non-causal" response is that the scattered field appears at a location within the scatterer earlier than the incident wave itself. Hence, locally, the process appears non-causal. Yet the nonlocal nature of the $\mathbb{T}$ matrix may be precisely what is needed to resolve this paradox. A $\mathbb{T}$ matrix isolates the response at any point $\mathbf{x}$ to the contributions from the wave incident at each point $\mathbf{x}'$ in the scatterer; each of which, individually, must be causal. Hence, not only should the $\mathbb{T}$ matrix be extensible to such scenarios; it may further resolve impediments that had previously stymied even simple sum rules in these fields. (Relatedly, wave scattering with any non-trivial/non-vacuum background has historically stymied sum rules, and this is another avenue of exploration with the $\mathbb{T}$ matrix.)

Finally, we speculate that the approach described here may even be extensible to quantum scattering. In the frequency domain, the key difference between quantum and classical scattering is the analytic structure of the governing equations. In classical wave equations, second derivatives in space are proportional to second derivatives in time, which lead to poles in the lower half of the complex-frequency plane and analyticity in the upper half. In quantum scattering, second derivatives in space are proportional to first derivatives in time, which leads to bounds states for negative real energies and branch cuts on the positive real axis. Our standard semicircular contours likely need to be replaced by "keyhole" contours[13], with the open question of whether there are meaningful sum rules that can be derived (perhaps dependent on bound-state properties, as in Levinson's theorem[59,60] for spherically symmetry potentials). If such sum rules could be derived, it is likely that an infinite-oscillator description could be used to identify fundamental limits for quantum scattering as well.

## Methods

### Domain monotonicity

In this section, we derive "domain montonicity" theorems for the matrices on the right-hand sides of the sum rules of Eqs. (4), (5). Domain monotonicity is trivial for the high-frequency sum rule, as the right-hand side is directly proportional to the identity matrix on $V$. Consider a domain $D$ that contains $V$. How can we compare the two identity matrices? We can embed the identity matrix on $V$ in a larger matrix on $D$, with zero elements for any spatial degrees of freedom in $D$ and not in $V$. Hence, by direct comparison, we will have

$$\mathbb{I}_V \leq \mathbb{I}_D, \tag{8}$$

proving that the high-frequency sum rule obeys domain monotonicity, implying that it can be converted to an inequality over any designable domain of interest.

Domain monotonicity for the low-frequency (static) sum rule is less obvious. Here, we generalize the arguments of ref.[22] to prove domain monotonicity. We need to prove that quantities of the form $\mathbf{x}^{\dagger}\mathbb{T}_{0,V}\mathbf{x}$ increase, for all $\mathbf{x} \neq 0$, when the domain $V$ increases (i.e., contains all points of its original domain, and a nonzero volume of points outside of its original domain), for a positive-semidefinite static susceptibility. (Gyrotropic materials, with a nonreciprocal pole at zero, are materials that do not have such susceptibilities[61].) We can interpret the multiplication of $\mathbb{T}$ with $\mathbf{x}$ as the polarization field induced by an "incident field" $\mathbf{x}$, and then multiplication on the left by $\mathbf{x}$ takes the overlap of that incident field with the polarization that it induces. Hence we will label our arbitrary vectors as $\mathbf{e}_{\text{inc}}$ instead of $\mathbf{x}$, for clarity in the mathematical relations to follow, though we impose no constraints on the "incident field" and indeed allow it to be an arbitrary

vector. In computing the response to such a vector, however, we can use a few important physical consequences of electromagnetism. In electrostatics, the fields (and $\mathbb{T}$ matrix) can be chosen to be real-valued, so that we can consider the objective as $\mathbf{x}^T \mathbb{T} \mathbf{x}$, without any conjugation.

We are interested in the quantity $F = \mathbf{e}_{inc}^T \mathbb{T} \mathbf{e}_{inc} = \mathbf{e}_{inc}^T \mathbf{p}$, and how it changes when the domain changes. We will consider only continuous, increasing changes in susceptibility: $\Delta\chi(\mathbf{x}) \geq 0$ everywhere. Hence a variation in $F$ can be written

$$\delta F = \mathbf{e}_{inc}^T \delta \mathbf{p}. \tag{9}$$

The polarization field is the solution of the volume (Lippmann–Schwinger) integral equation:

$$(\mathbb{G}_0 + \xi)\mathbf{p} = -\mathbf{e}_{inc}, \tag{10}$$

where $\mathbb{G}_0$ is the background (vacuum) Green's function operator, $\xi = -\chi^{-1}$, and $\mathbf{e}_{inc}$ is the incident field. The variation in $\mathbf{p}$ can be found by taking the variation of Eq. (10), which is: $(\mathbb{G}_0 + \xi)\delta\mathbf{p} + (\delta\xi)\mathbf{p} = 0$. Solving for $\delta\mathbf{p}$:

$$\delta\mathbf{p} = -(\mathbb{G}_0 + \xi)^{-1}(\delta\xi)\mathbf{p}. \tag{11}$$

Inserting this variation into the objective gives

$$\begin{aligned} \delta F &= -\mathbf{e}_{inc}^T (\mathbb{G}_0 + \xi)^{-1}(\delta\xi)\mathbf{p} \\ &= \mathbf{p}^T (\delta\xi)\mathbf{p}. \end{aligned} \tag{12}$$

Finally, from the equation $\xi = -\chi^{-1}$, we have $\delta\xi = \chi^{-1}(\delta\chi)\chi^{-1}$, so that

$$\delta F = \mathbf{e}^T (\delta\chi)\mathbf{e}, \tag{13}$$

which is nonnegative for any positive semidefinite $\delta\chi$. Hence we have shown that

$$\mathbf{e}_{inc}^\dagger \delta\mathbb{T} \mathbf{e}_{inc} \geq 0 \tag{14}$$

for any increases in the domain size or shape; since this is true for any vector $\mathbf{e}_{inc}$, then variations in the electrostatic $\mathbb{T}$ matrix must themselves be monotonic. This means that given a scatterer $\Omega_1$ of any size and shape whose static $\mathbb{T}$ matrix is $\mathbb{T}^{(1)}(\omega = 0)$, any other scatterer $\Omega_2$ whose volume encloses that of $\Omega_1$ must have a $\mathbb{T}^{(2)}(\omega = 0)$ no smaller than $\mathbb{T}^{(1)}(\omega = 0)$, i.e.:

$$\mathbb{T}^{(2)}(\omega = 0) \geq \mathbb{T}^{(1)}(\omega = 0), \tag{15}$$

when the scatterer domain $\Omega_2$ entirely encloses the scatter domain $\Omega_1$.

## Derivation of the NFRHT bound

To investigate radiative heat transfer from object 1 (bottom) to object 2 (top), we first break down the problem to power integrations at every frequency. The power flowing in the positive $z$ direction across the middle separating plane (perpendicular to $z$) between the two objects is:

$$S(\omega) = \frac{1}{2} \mathrm{Re} \int dS \left[ \left( E_x^J(\mathbf{r}) \right)^* H_y^J(\mathbf{r}) - \left( E_y^J(\mathbf{r}) \right)^* H_x^J(\mathbf{r}) \right], \tag{16}$$

where the superscripts denote the current sources in the bottom object, whose amplitudes are dictated by the fluctuation-dissipation theorem:

$$\langle J_i^*(\omega, \mathbf{r}_v) J_j(\omega', \mathbf{r}_v') \rangle = Z(\omega, T)\delta(\omega - \omega')\delta(\mathbf{r}_v - \mathbf{r}_v')\delta_{ij}, \tag{17}$$

where $Z(\omega, T) = \frac{4\varepsilon_0 \omega}{\pi} \mathrm{Im}\chi_1(\omega)\Theta(\omega, T)$, the susceptibility of the lower body is $\chi_1(\omega) = \frac{\varepsilon_1(\omega)}{\varepsilon_0} - 1$, and $\Theta(\omega, T)$ is the Planck distribution, $\Theta(\omega, T) = \hbar\omega / (e^{\frac{\hbar\omega}{k_B T}} - 1)$. The subscripts in $\mathbf{r}_v$ indicate the position vector lies in the volume of the emitter 1. Then the field correlations in Eq. (16) can be expressed in terms of the Green's functions $G^{EJ}(\mathbf{r}, \mathbf{r}_v')$ and $G^{HJ}(\mathbf{r}, \mathbf{r}_v')$ applied to the thermal source correlations in Eq. (17).

Our bound will not distinguish between the $x$ and $y$ directions (which are symmetric in the bounding domain, even though of course they are not for many allowable patterns), in which case the upper bounds on either of the two terms in power integration in Eq. (16) are identical: $\mathrm{Max}[\mathrm{Re} \int dS(E_x^J(\mathbf{r}))^* H_y^J(\mathbf{r})] = \mathrm{Max}[-\mathrm{Re} \int dS(E_y^J(\mathbf{r}))^* H_x^J(\mathbf{r})]$. Hence the maximum flux $S(\omega, T)$ equals the maximum of the function

$$F^{\mathrm{RHT}}(\omega, T) \equiv \mathrm{Re} \int dS \left( E_x^J(\mathbf{r}) \right)^* H_y^J(\mathbf{r}). \tag{18}$$

We use reciprocity to transfer the flux evaluation of Eq. (18) on the surface $S$ from sources in $V$ to a field evaluation in $V$ from sources on $S$. The background Green's functions are reciprocal, i.e., $G_{ik}^{EJ}(\mathbf{r}, \mathbf{r}_v') = G_{ki}^{EJ}(\mathbf{r}_v', \mathbf{r})$, and $G_{ik}^{HJ}(\mathbf{r}, \mathbf{r}_v') = -G_{ki}^{EM}(\mathbf{r}_v', \mathbf{r})$, so we can equate the fields at $\mathbf{r}$ produced by sources at $\mathbf{r}_v$ with fields at $\mathbf{r}_v$ produced by sources at $\mathbf{r}$. In light of the correlations for currents sources inside the volume, Eq. (17), we can define the correlations for *reciprocal* current sources on the middle flux plane as

$$\langle J_x^*(\omega, \mathbf{r}) M_y(\omega', \mathbf{r}') \rangle \equiv \omega\delta(\omega - \omega')\delta(\mathbf{r} - \mathbf{r}'). \tag{19}$$

The amplitude $\omega$ is chosen so that $\left( E_{inc}^{J_x}(\mathbf{r}_v) \right)^* E_{inc}^{M_y}(\mathbf{r}_v)$ is independent of frequency, which will be important later. Simple insertion of the Green's functions into Eq. (18) and the usage of reciprocity and Eq. (19) leads to a volume-field expression for $F^{\mathrm{FHT}}$:

$$F^{\mathrm{RHT}}(\omega, T) = \frac{Z(\omega, T)}{\omega} \mathrm{Re} \int_{V_S} dV \left( \mathbf{E}^{J_x}(\mathbf{r}_v) \right)^* \mathbf{E}^{M_y}(\mathbf{r}_v) \tag{20}$$

where $V_S$ is exclusively the source volume. Equation (20) represents the total flux from an infinite plane of sources between the infinite bodies. An upper bound on this flux is given by the upper bound on the flux generated by a single set of point sources at a given position on the separating plane, multiplied by the (infinite) area of the plane. This allows us to easily switch to the quantity of interest in large-area NFRHT: the per-area radiative heat transfer, which is bounded above by the maximum flux from a single set of sources at a single position on the separating plane. This also resolves a second possible difficulty: how to represent the $\mathbb{T}$ matrix for infinite, extended structures? For point sources in the near field, there is no issue: the fields decay sufficiently quickly that the response is guaranteed to be well-behaved. (Intuitively, one can imagine substituting large but finite-sized structures at this stage, and later taking the limit as size goes to infinity. The rapid field decay ensures that the subsequent integrals converge, even in the infinite-size limit.)

We switch to vector notation now, using the notation of lowercase letters without the subscript $v$ to represent field vectors on the domain of both objects. For example, the volume integral over the lower body in Eq. (20) becomes $(\mathbf{e}_v^{J_x})^\dagger \mathbb{O} \mathbf{e}_v^{M_y}$, where $\mathbb{O}$ has ones on its diagonal in the lower (source) volume and zeros everywhere else. We can write this integral out in terms of the $\mathbb{T}$ matrix:

$$\begin{aligned} \mathrm{Re}\left[ (\mathbf{e}^{J_x})^\dagger \mathbb{O} \mathbf{e}^{M_y} \right] &= \frac{1}{\varepsilon_0^2 |\chi|^2} \mathrm{Re}\left( \left( \mathbf{e}_{inc}^{J_x} \right)^\dagger \mathbb{T}^\dagger \mathbb{O} \mathbb{T} \mathbf{e}_{inc}^{M_y} \right) \\ &= \frac{1}{\varepsilon_0^2 |\chi|^2} \mathrm{Tr}\left( \mathbb{T}^\dagger \mathbb{O} \mathbb{T} \; \mathbb{E} \right). \end{aligned} \tag{21}$$

Notice both $\mathbb{T}$ matrix and $\mathbf{e}_{inc}$ vectors are defined on the domain of both the top and bottom bodies. In Eq. (21) we defined the function

$\mathbb{E} = \mathrm{Re}\left(\boldsymbol{e}_{\mathrm{inc}}^{M_y}\left(\boldsymbol{e}_{\mathrm{inc}}^{J_x}\right)^{\dagger}\right)$ which is a rank-2 matrix and can be decomposed into one positive eigenvalue term and one negative eigenvalue term:

$$\mathbb{E} = \lambda_1 q_1 q_1^{\dagger} + \lambda_2 q_2 q_2^{\dagger}, \tag{22}$$

with eigenvector $q_{1,2}$ and eigenvalues $\lambda_{1,2}$ given by

$$q_{1,2} = \frac{\boldsymbol{e}_{\mathrm{inc}}^{J_x}}{\sqrt{2}|\boldsymbol{e}_{\mathrm{inc}}^{J_x}|} \pm \frac{\boldsymbol{e}_{\mathrm{inc}}^{M_y}}{\sqrt{2}|\boldsymbol{e}_{\mathrm{inc}}^{M_y}|}, \tag{23}$$

$$\lambda_{1,2} = \pm \frac{|\boldsymbol{e}_{\mathrm{inc}}^{J_x}||\boldsymbol{e}_{\mathrm{inc}}^{M_y}|}{2} = \pm \frac{1}{\varepsilon_0} \frac{3.45 \times 10^{16}}{\left(d \times 10^9\right)^2}. \tag{24}$$

One can now see that our choice of source amplitudes in Eq. (19) leads to frequency-independent eigenvalues of $\mathbb{E}$.

To bound the expression of Eq. (21), we will relax it in a few ways. (Interestingly, intensive numerical optimizations using manifold-optimization techniques[62,63] directly on Eq. (21) lead to the same upper limits that we derive below, suggesting that these "relaxations" are minimal and do not loosen the analysis given the constraints that we use, such as sum rules.) First, the $\mathbb{E}$ matrix defined by the two renormalized incident fields has one positive and one negative eigenvalue, per Eq. (22). Physically, we can interpret the negative sign of the second eigenvalue via the power expression of Eq. (21) containing $\mathbb{E}$, as the difference in powers absorbed for the two renormalized incident fields. This is of course bounded above by the absorption of only the first incident field, dropping the subtracted term, leaving only the contribution of the single positive eigenvalue of $\mathbb{E}$. Thus we have:

$$\mathrm{Tr}\left[\mathbb{T}^{\dagger} \mathbb{O} \mathbb{T} \mathbb{E}\right] \leq \mathrm{Tr}\left[\mathbb{T}^{\dagger} \mathbb{O} \mathbb{T} \, \lambda_1 q_1 q_1^{\dagger}\right]. \tag{25}$$

Next, we note that $\mathbb{O}$ indicates absorption only in the lower body; of course this quantity is bounded above by the total absorption in both bodies. This is represented mathematically as the constraint that $\mathbb{O} \leq \mathbb{I}$, which implies:

$$\lambda_1 \left(q_1^{\dagger} \mathbb{T}^{\dagger} \mathbb{O} \mathbb{T} q_1\right) \leq \lambda_1 \left(q_1^{\dagger} \mathbb{T}^{\dagger} \mathbb{T} q_1\right). \tag{26}$$

Finally, the absorption in both bodies is less than the net extinction of the two bodies (their far-field scattered powers are positive, and essentially zero in the near-field case, so that this relaxation is negligible). We can use a generalized "optical theorem" constraint to bound this quadratic absorption-like quantity with a linear extinction-like quantity. The idea is that absorption must be smaller than extinction: $P_{\mathrm{abs}} \leq P_{\mathrm{ext}}$. Absorption is given in terms of $\mathbb{T}$-matrix by $P_{\mathrm{abs}} = \frac{\omega}{2} \mathrm{Im}(\boldsymbol{e}^{\dagger} \mathbf{p}) = \frac{\omega}{2\varepsilon_0} \frac{\mathrm{Im}\chi}{|\chi|^2} \boldsymbol{e}_{\mathrm{inc}}^{\dagger} \mathbb{T}^{\dagger} \mathbb{T} \boldsymbol{e}_{\mathrm{inc}}$. Similarly extinction is given by $P_{\mathrm{ext}} = \frac{\omega}{2} \mathrm{Im}(\boldsymbol{e}_{\mathrm{inc}}^{\dagger} \mathbf{p}) = \frac{\omega}{2} \boldsymbol{e}_{\mathrm{inc}}^{\dagger} (\mathrm{Im}\,\mathbb{T}) \boldsymbol{e}_{\mathrm{inc}}$. Thus the "optical theorem" condition implies that for any $\mathbb{T}$ matrix,

$$\frac{\mathrm{Im}\chi}{\varepsilon_0 |\chi|^2} \mathbb{T}^{\dagger} \mathbb{T} \leq \mathrm{Im}\,\mathbb{T}. \tag{27}$$

Hence we can write

$$\lambda_1 \left(q_1^{\dagger} \mathbb{T}^{\dagger} \mathbb{T} q_1\right) \leq \lambda_1 \frac{\varepsilon_0 |\chi|^2}{\mathrm{Im}\chi} q_1^{\dagger} (\mathrm{Im}\,\mathbb{T}) q_1, \tag{28}$$

without introducing much relaxation. We can now rewrite Eq. (20) as

$$F^{\mathrm{RHT}}(\omega,T) \leq \frac{Z(\omega,T)}{\omega} \frac{1}{\varepsilon_0^2 |\chi|^2} \lambda_1 \frac{\varepsilon_0 |\chi|^2}{\mathrm{Im}\chi} q_1^{\dagger} (\mathrm{Im}\,\mathbb{T}) q_1 \tag{29}$$

$$= \frac{4\lambda_1}{\pi} \Theta(\omega,T) q_1^{\dagger} (\mathrm{Im}\,\mathbb{T}) q_1. \tag{30}$$

Surprisingly, the various transformations to this point have removed all explicit dependencies on material susceptibility $\chi_{1,2}(\omega)$, with the only implicit dependence embedded in $\mathrm{Im}\,\mathbb{T}$. We will now focus on the upper bound for HTC, and the upper bound for RHT can be found by taking similar steps. To switch from the RHT to HTC bound computation, we just need to take the temperature derivative of the last expression to get

$$F^{\mathrm{HTC}}(\omega,T) \leq \frac{4\lambda_1}{\pi} \frac{\partial \Theta(\omega,T)}{\partial T} q_1^{\dagger} (\mathrm{Im}\,\mathbb{T}) q_1. \tag{31}$$

In our oscillator representation, we know that $\omega \mathrm{Im}\,\mathbb{T}(\omega)$ is exactly the real-symmetric positive-semidefinite matrix $\mathbb{X}(\omega)$, which must satisfy the low-frequency sum rule $\int_0^{\infty} \mathbb{X}(\omega_i)/\omega_i^2 \, \mathrm{d}\omega \leq \alpha \mathbb{I}_D$. (The nonreciprocal part of $\omega \mathrm{Im}\,\mathbb{T}(\omega)$ cannot contribute, as the NFRHT objective is symmetric around $\omega = 0$, so that it can be written as the linear combination of positive-frequency contributions and their negative-frequency counterparts. The positive- and negative-frequency contributions cancel for the nonreciprocal part due to its anti-symmetry in frequency.) We renormalize $\mathbb{X}$ to simplify the sum rule: $\mathbb{X}(\omega_i) \to \alpha(\pi/2)\omega_i^2 \mathbb{X}(\omega_i)$, so that $\int_0^{\infty} \mathbb{X}(\omega_i) \leq \mathbb{I}_D$. In terms of $\mathbb{X}(\omega_i)$, the total frequency-integral HTC is

$$\mathrm{HTC} \leq 2\varepsilon_0 \alpha \lambda_1 \mathrm{Tr}\left[q_1 q_1^{\dagger} \int_0^{\infty} \mathbb{X}(\omega)\left(\omega \frac{\partial \Theta(\omega,T)}{\partial T}\right) \mathrm{d}\omega\right]. \tag{32}$$

The optimization of Eq. (32), subject to the passivity constraint ($\mathbb{X}(\omega) \geq 0$) and the sum-rule constraint ($\int_0^{\infty} \mathbb{X}(\omega) \, \mathrm{d}\omega \leq \mathbb{I}_D$) is actually simple, thanks to the structure of the objective and representation. We form a basis $\mathbb{Q}$ whose first column is $q_1$, with all other columns orthogonal to $q_1$. If we write at every frequency $\mathbb{X}(\omega) = \mathbb{Q}\mathbb{X}'(\omega)\mathbb{Q}^{\dagger}$, then $\mathrm{Tr}\left[q_1 q_1^{\dagger} \mathbb{X}(\omega)\right] = q_1^{\dagger} \mathbb{Q}\mathbb{X}'(\omega)\mathbb{Q}^{\dagger} q_1 = (\mathbb{X}'(\omega))_{11}$. Hence only the $(1,1)$ element of $\mathbb{X}'(\omega)$ contributes to the objective (due to the rank-one nature of the excitation). The positive semidefinite property as well as the sum rule for $\mathbb{X}(\omega)$ are equivalent for $\mathbb{X}'(\omega)$ (as the transformation was unitary). Hence we can rewrite the HTC bound as:

$$\mathrm{HTC} \leq 2\varepsilon_0 \alpha \lambda_1 \int_0^{\infty} (\mathbb{X}'(\omega))_{11}\left(\omega \frac{\partial \Theta(\omega,T)}{\partial T}\right) \mathrm{d}\omega, \tag{33}$$

subject to the constraints $\mathbb{X}'_{11}(\omega) \geq 0$ and $\int_0^{\infty} \mathbb{X}'_{11}(\omega) \, \mathrm{d}\omega \leq 1$. The maximization of an inner product subject to a "probability simplex" constraint[64] has a simple solution: concentrate all of the response into the single degree of freedom where the objective vector is maximized. In particular, in this case, the optimal $\mathbb{X}'_{11}$ is a delta function with unit amplitude at the frequency where $\omega \frac{\partial \Theta(\omega,T)}{\partial T}$ is maximized. A simple calculation shows that this occurs for

$$\omega_{\mathrm{opt}} = \frac{2.57 k_B T}{\hbar} = \frac{x_{\mathrm{opt}} k_B T}{\hbar}, \tag{34}$$

which is exactly the near-field Wien frequency that we found from HTC optimization for planar, unpatterned geometries[43]. In terms of the dimensionless variable $x_{\mathrm{opt}} = 2.57$, the HTC bound is:

$$\mathrm{HTC} \leq 2\varepsilon_0 \alpha \lambda_1 \left(\omega_{\mathrm{opt}} \frac{\partial \Theta(\omega_{\mathrm{opt}},T)}{\partial T}\right) \tag{35}$$

$$= 2\varepsilon_0 \alpha \lambda_1 \frac{k_B^2 T}{\hbar} \frac{x_{\mathrm{opt}}^3 e^{x_{\mathrm{opt}}}}{(e^{x_{\mathrm{opt}}} - 1)^2}. \tag{36}$$

Inserting the numerical prefactors, we arrive at the final bound:

$$\text{HTC} \leq 0.11\alpha \frac{k_B^2}{\hbar} \frac{T}{d^2} = \beta \frac{T}{d^2}, \tag{37}$$

where $\beta = 3.8 \times 10^5 \, \text{Wnm}^2/\text{m}^2/\text{K}^2$. For $T = 300\,\text{K}$ and $d = 10\,\text{nm}$, $\text{HTC} \leq 1.1 \times 10^6 \, \text{W/m}^2/\text{K}$, which is 5X the optimal planar performance. Hence this theoretical framework offers a close prediction to the best known designs, it predicts the optimal resonance frequency where the oscillator-strength should be concentrated, and it explains why previous material-dependent predictions were incorrect.

## Data availability

The datasets generated in this study are available at https://github.com/PhotonDesign/ScatteringOscillatorsResults.

## Code availability

The simulation code used in this study is available at https://github.com/PhotonDesign/ScatteringOscillatorsResults.

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

## Acknowledgements

The authors thank Wenjin Xue and Hanwen Zhang for providing the fast integral solver for computing the $\mathbb{T}(\omega)$ matrix of the elliptical scatterer. This work was supported by: Air Force Office of Scientific Research Grant No. FA9550-22-1-0393 (L.Z., O.D.M., general $\mathbb{T}$-matrix theory), Army Research Office Grant No. W911NF-19-1-0279 (L.Z., O.D.M., near-field heat transfer analysis), Air Force Office of Scientific Research Grant No. FA9550-22-1-0204 (F.M.), and Office of Naval Research Grant No. N00014-22-1-2486 (F.M.).

## Author contributions

O.D.M. and F.M. conceived the initial idea. L.Z. and O.D.M. developed the formalism and example demonstrations. All authors discussed the technical aspects and features as well as possible extensions. All authors wrote the manuscript.

## Competing interests

The authors declare no competing interests.
