## [Peer Review File · Nature Communications]

All electromagnetic scattering bodies are matrix-valued oscillatorsREVIEWER COMMENTS

Reviewer #1 (Remarks to the Author):

In this paper, the authors cover two main points. First, they justify a representation of the T matrix as a sum of matrix-valued oscillators with specific constraints. Second, they apply this representation to derive an analytical limit to the heat transfer coefficient in a particular problem setting. Overall the work is interesting and the supplemental materials cover many of the technical details related to both points.

I have a few criticisms of the paper mainly focused on the suggested novelty and impact of the “key result” as it is presented in (5) and Fig. 1. On a more general note, I have concerns with how much of the paper is contained in the SM itself, leaving the communication to serve as an overly-brief abstract for a dense body of thoughtful theoretical work.

First, the three points (KK, passivity, sum rule) described as the “key result” around and including (5) are, on their own, very well-known. Their extension to non-reciprocal systems in SM Sec. IV is nice, but those cases aren’t discussed in any significant detail in the manuscript. I also find it surprising that there are virtually no references in that section of the SM, given the ubiquitous use of KK relations in the study of all sorts of optical, acoustic, material, and network problems over the last several decades. To reiterate, the delta-function basis in (4) and the subsequent model in (5) doesn’t leverage any of the new results related to non-reciprocal systems derived in the SM.

The example data in Fig. 1 provide little additional perspectives on the “main result” of (5).
- Panel C shows diagonal elements of the T matrix. Is it correct that this is a reciprocal system? If so, why are both real and imaginary parts shown when the axis label reads $\text{Im } T$?
The fact that some of the traces exhibit consecutive oscillator-like responses is not surprising, given that the quantities being plotted are analogous to the circuit admittances seen by the incident field at each location. If the bottom panel were taken in isolation, it could be interpreted as clearly defined cascade of oscillator circuits. The central panel taken in isolation would not lead to that conclusion. The caption text stating that this is a “new type of matrix-valued, nonlocal scattering oscillator” overlooks the prevalence and well-

studied nature of these responses in systems as simple as a lossy Fabry-Perot etalon.

- Panel D shows the matrices T_i at three frequencies, which I assume are taken directly from the computed T matrix at the selected frequencies via (4). In my opinion, these pictures and their supporting text provide little quantitative or qualitative information to the reader.

What data or patterns in those pictures “showcase the extent to which passivity and causality restrict the possible form of the scattering T matrix”?

- In Panel E, the positivity of the eigenvalues of the $\text{Im } T$ matrix is demonstrated. The PSD-nature of this matrix (which represents many things, such as the admittance parameters of a circuit network) is well-known throughout electromagnetics, from optics to microwave circuits. Its applications range from the verification of measurement data to constraining optimization problems. By showing that $\text{Im } T$ has positive eigenvalues, this panel serves only to verify that the numerical modeling accurately enforces the system’s passivity.

In summary, Fig. 1 does not demonstrate any new information. The statement that it helps in the paper “uncovering this structure [of the T matrix]” (referring to the results near (5)) is a very big stretch.

The example calculation demonstrating the limits on NFRHT is very interesting, and I appreciate the details given in the SM. However, the manuscript’s single page and figure devoted to this complex derivation, its results, and subtle interpretations doesn’t do the work justice. The 10 pages of SM on this topic contain a considerable amount of technical work that should be the subject of its own critical review.

The final section of the paper is an excellent forward-thinking discussion of open problems in electromagnetics, acoustics, and quantum physics. I agree that the study and application of the T matrix (or its many other names from various integral equations, SM Sec. II) will likely be a fruitful path forward in many areas.

In conclusion, the 32 pages of scientific content of the SM is very interesting, however I believe that the paper itself is an overly condensed abstract of this very broad work. The example data shown in Fig. 1 do little to support the “uncovering this structure [of the T matrix]” and the NFRHT summary is, realistically, far too dense to comprehend without a

detailed study of 10 pages of SM.

I would strongly suggest that the authors consider dividing this work up into more traditional journal articles on their work regarding the structure of the T matrix for non-reciprocal systems (SM Secs. I-VI) and the application of those results to NFRHT (SM Secs. VIII-IX). This would make the technical content of the work more accessible and would allow for more focused scientific reviews.

Reviewer #2 (Remarks to the Author):

The work by Zhang, Monticone and Miller reports on a new approach to derive fundamental bounds on the performance of scattering nanophotonic structures. The approach is based on the new realization that the “T-matrix” of a scatterer can be decomposed as a sum of T-matrices that each have the frequency dependence of a Lorentz oscillator, and bounded by causality and passivity.

In my view, the paper is very interesting, and the application to the near-field radiative heat transfer problem shows that the oscillator decomposition indeed allows to derive fundamental bounds. At the same time, I believe that the paper may not be suited for Nature Communications, at least in its present form, for the following reasons:

- All the “meat” of the paper is actually in the supplement. Thus the paper is in fact more a manual to the actual description of the theory, and in itself not very clear. This means the paper does not really do justice to its content.
- To illustrate how this paper is “not very clear”: For instance, surrounding Eq. (4) and (5) and Figure 1, the authors state that “a discrete set of frequencies ω_i ” is chosen but they never state how this choice is made, given a problem. From the supplement it appears that the ω_i are chosen simply as an equidistant discretization of the frequency axis, but instead when discussing Fig 1c on page 3 (second column), the reader is supposed to recognize oscillators. Indeed, one sees of order 8 resonances in the spectrum. However, as these actually have a distinct frequency width, the relation between these resonances, how

these oscillators could be lossless, and how they differ from the signatures of the QNMs from literature, is unclear.

I believe that for this paper to resonate with the reader, it should be made much clearer how the authors construct the oscillators ω_i for the given scattering problem.

Supposing that they have access to the calculated full T-matrix versus frequency, then how would they go about defining the oscillators ω_i , and using the resulting construction to their advantage? The authors claim a distinction with the field of QNMs, but at least in the QNM field there are recipes for how to find them in the first place, and how to use QNMs to calculate observables. For the example of figure 1, to highlight the claimed distinction with QNMs, it would be helpful to highlight both the real-valued oscillators at ω_i quantitatively, and any QNMs that readers may wish to read into Fig. 1c instead.

Without such pointers I believe it will be impossible for an interested reader to take up this result, and apply it to a problem of their own choosing. How to apply the method to derive a bound for a problem of choice other than the example, or if the method could even be used for design, is not clear from the paper.

- The fact that the oscillators are real-valued is claimed as crucial by the authors. Yet the authors introduce loss as $i\omega\gamma$, (Eq. 5), and do not discuss how to deal with the problems that surely must arise in applying the limiting procedure $\gamma \rightarrow 0$. Here I note that also the supplement on this point is surprisingly brief – so much so that the fact that a limit must be taken is not even written down let alone discussed for the NFRHT example.

- Finally I note that the notion of T-matrix presupposes that one makes a separation between scatterer, and background of which one must know the Green function. I find that the paper is unclear about how this distinction must generally be made, and for instance how one deals with resultant problematic properties of G (e.g., guided modes in planar systems), or in the scatterer (e.g., Equation 17 of the supplement assumes that the scatterer has a finite extent – but it appears that the NFRHT is infinitely spatially extended).

Report of Zhang et al. “All electromagnetic scattering bodies are matrix-valued oscillators”

This paper presents an interesting theoretical analysis of electromagnetic scattering. It proposes a new mathematical representation of the T -matrix that incorporates the principles of causality and passivity. The authors use this representation to uncover hidden resonant patterns in scattered fields. As an application, the authors develop a general theory of maximum radiative heat transfer in the near field that provides a better theoretical bound.

I enjoyed reading this paper as it contains many novel thoughts. It provides a different viewpoint on wave scattering from the coupled-mode theory or quasi-normal mode expansion. This new viewpoint is particularly useful in treating physical problems involving absorption and emission. The main results of this paper can be re-derived as follows:

The T -matrix is causal and thus satisfies the Kramers-Kronig relation:

$$\operatorname{Re} T(\omega) = \frac{1}{\pi\omega} * \operatorname{Im} T(\omega) \quad (\text{a})$$

where $*$ denotes convolution. Therefore,

$$\begin{aligned} T(\omega) &= \operatorname{Re} T(\omega) + i \operatorname{Im} T(\omega) \\ &= \int_{-\infty}^{\infty} \left[\frac{1}{\pi(\omega' - \omega)} + i\delta(\omega' - \omega) \right] \operatorname{Im} T(\omega') d\omega' \\ &= \lim_{\varepsilon \rightarrow 0} \int_{-\infty}^{\infty} \frac{1}{\pi(\omega' - \omega - i\varepsilon)} \operatorname{Im} T(\omega') d\omega' \end{aligned} \quad (\text{b})$$

For a reciprocal system, $\operatorname{Im} T(-\omega') = -\operatorname{Im} T(\omega')$, then Eq. (b) becomes:

$$T(\omega) = \lim_{\gamma \rightarrow 0} \int_0^{\infty} \frac{2\omega'}{\pi(\omega'^2 - \omega^2 - i\gamma\omega)} \operatorname{Im} T(\omega') d\omega' \quad (\text{c})$$

where we define $\gamma = 2\varepsilon$. This is Eq. (5) in the paper once one substitutes Eq. (4) into Eq. (c). If the system is passive, then $\operatorname{Im} T(\omega')$ is positive semidefinite.

Now the mathematical content is clear. Causality reduces the degree of freedom for $T(\omega)$ by half: Knowing $\operatorname{Im} T(\omega)$ is sufficient to reconstruct $T(\omega)$. Moreover, the integral representation of $T(\omega)$ has a Lorentzian-like weight factor. These observations lead to the title of this paper: All electromagnetic scattering bodies are matrix-valued oscillators.

Eq. (c) is a new mathematical representation of the T -matrix that incorporates causality and passivity. Whether it is useful to physicists depends on, of course, what new physical

insights this mathematical representation can provide. Revealing these physical implications is the main contribution of this paper. The authors have successfully illustrated how this mathematical representation can reveal hidden patterns in wave scattering using concrete physical examples (Fig. 1). They also demonstrate how this representation can shed light on an open question on the theoretical limit of near-field radiative heat transfer (Fig. 2). The new theoretical bound is much closer to the state-of-the-art performance.

As such, I think this work is a solid and important theoretical work that merits its publication in Nature Communications. It should be of interest to a broad audience in the physics community.

Nonetheless, I do have a few comments for the authors to improve their manuscript.

1. The authors have focused on the T -matrix. Another fundamental object in wave scattering is the S -matrix. I was wondering whether the authors' results on the T -matrix have any implications on the S -matrix. A discussion on this would be useful.
2. The meaning of Eq. (4) needs to be clarified. Strictly speaking, the left is a matrix-valued function, while the right is a distribution. It is not clear to me how equality can be established here. In addition, there is a typo in Eq. (4): it should be $\delta(\omega - \omega_i)$.
3. The authors have written the conditions $\sum_i \mathbb{T}_i = \mathbb{I}$ and $\sum_i \mathbb{T}_i \leq \mathbb{I}$ simultaneously. It seems to me that Eq. (3) implies $\sum_i \mathbb{T}_i = \mathbb{I}$. Where does the weaker condition $\sum_i \mathbb{T}_i \leq \mathbb{I}$ come from? Why do we need it?
4. Fig. 1c and 1d: The authors have only presented the results on the diagonal elements of the T -matrix. What about the off-diagonal elements of the T -matrix? Do they exhibit similar behaviors?

Overall, I think the authors have done an excellent job of presenting their new mathematical representation of the T -matrix and demonstrating its usefulness in wave scattering. I believe that addressing the above comments will further improve the manuscript.

All reviewer comments are included below in black text; our responses are in blue text.

We thank all reviewers for their time and their many thoughtful suggestions.

Reviewer #1:

1. In this paper, the authors cover two main points. First, they justify a representation of the T matrix as a sum of matrix-valued oscillators with specific constraints. Second, they apply this representation to derive an analytical limit to the heat transfer coefficient in a particular problem setting. Overall the work is interesting and the supplemental materials cover many of the technical details related to both points.

We thank the reviewer for their time and are glad that they find the results are interesting.

2. I have a few criticisms of the paper mainly focused on the suggested novelty and impact of the “key result” as it is presented in (5) and Fig. 1. On a more general note, I have concerns with how much of the paper is contained in the SM itself, leaving the communication to serve as an overly-brief abstract for a dense body of thoughtful theoretical work.

First, the three points (KK, passivity, sum rule) described as the “key result” around and including (5) are, on their own, very well-known. Their extension to non-reciprocal systems in SM Sec. IV is nice, but those cases aren’t discussed in any significant detail in the manuscript.

We certainly agree that KK, passivity, and sum rules, as general ideas, are well-known! However, their consequences for one special scattering matrix – the volume T matrix, or equivalently a volume admittance matrix – are not well-known. First, the sum rules that we derive for the T matrix (Eqs. 4,5 in the revised manuscript) do appear to be entirely new. Second, what is particularly unique is the *tandem* combination of the KK relation, the passivity constraint, and the new sum rules. Only the volume T matrix appears to have this combination; previous work primarily emphasized representations, without bounds, likely because for most scattering matrices (e.g., the surface scattering matrix, the surface impedance matrix, the volume scattering matrix, etc.) one *cannot* combine them into a simple mathematical form with constraints that comprise a bounded, convex set. In our revised manuscript, we try to consistently make this point more clearly throughout.

Previous efforts have fallen short of combining all three of these ideas for a wave-based scattering matrix. We can review various threads that have been pursued:

1. In much of the early foundational work on passive linear systems (now reviewed in Sec. IIIA of the SM), it was recognized that immittance (impedance/admittance) matrices have positive semidefinite Hermitian parts in the UHP, and some connections to Herglotz—Nevanlinna relations that are similar to KK relations. But when KK relations themselves were given, they were always *entrywise* KK relations (cf., e.g., Eq. (42) of the review paper: A. Srivastava, “Causality and passivity: From electromagnetism and network theory to metamaterials,” *Mechanics of Materials* 154, 103710 (2021)). The problem with entrywise KK relations is that the off-diagonal entries of the relevant matrices do not have any direct positivity constraints; passivity

instead enforces constraints on the Hermitian/anti-Hermitian parts of the matrices. Hence an entrywise KK relation cannot be combined with passivity for a representation whose degrees of freedom reside in a bounded, convex set.

2. For special scattering quantities such as extinction or local density of states, the power objective can be written as the real or imaginary part of a scattering amplitude (not its magnitude squared). This has led to quite useful sum rules (e.g. Refs. A-D). But for any scattering problem with any objective other than these special ones, those sum rules offer no utility. Here, by zeroing in on the T matrix itself, we construct a representation that should be useful for a broad array of objectives (such as radiative heat transfer).
3. The mathematical structure of passivity constraints has been a bounty for identifying fundamental limits. Indeed, our groups (Miller and Monticone), and the groups of Alejandro Rodriguez, Mats Gustafsson and collaborators, Jelena Vuckovic, Andrea Alu, and many others have been using such constraints for a decade to identify limits to electromagnetic response. (A recent review is given by Rodriguez et al. "Physical limits in electromagnetism," *Nature Reviews Physics* 4.8 (2022): 543-559.) Yet passivity by itself only enables bounds at a single frequency. Those techniques had not been combined with a spectral representation of scattering matrices that could lead to broadband bounds.

- A. Gordon, R. G. "Three sum rules for total optical absorption cross sections." *The Journal of Chemical Physics* 38.7 (1963): 1724-1729.
- B. Purcell, E. M. "On the absorption and emission of light by interstellar grains." *Astrophysical Journal*, vol. 158, p. 433 158 (1969): 433.
- C. Sohl, Christian, Mats Gustafsson, and Gerhard Kristensson. "Physical limitations on broadband scattering by heterogeneous obstacles." *Journal of Physics A: Mathematical and Theoretical* 40.36 (2007): 11165.
- D. Shim, Hyunki, et al. "Fundamental limits to near-field optical response over any bandwidth." *Physical Review X* 9.1 (2019): 011043.

3. I also find it surprising that there are virtually no references in that section of the SM, given the ubiquitous use of KK relations in the study of all sorts of optical, acoustic, material, and network problems over the last several decades.

We thank the reviewer for this suggestion, and agree with it. We now include many references at the appropriate places. We also now describe the key foundational results of passive linear system theory – including KK and KK-like representations – in the new Sec. IIIA of the SM.

4. To reiterate, the delta-function basis in (4) and the subsequent model in (5) doesn't leverage any of the new results related to non-reciprocal systems derived in the SM.

Indeed, our emphasis in this manuscript is on reciprocal systems. As we contend above, our T-matrix representation is new and useful *even for reciprocal systems*. The extra degrees of freedom in nonreciprocal systems offer no benefit for any objective (such as radiative heat transfer) that is symmetric around the imaginary frequency axis. Such symmetry is present in most objectives (power flow, momentum transfer, etc., as long as they are not explicitly nonreciprocal); for most optical systems, reciprocal designs can be optimal, and our reciprocal representation theorem should be useful.

We agree that the nonreciprocal results are also new and interesting, especially in the context of much current interest in breaking reciprocity, and we now include them in our key

representation result in the main text. (We used a more compact derivation to add this to the main text.) We do plan to explore novel applications of the nonreciprocal T-matrix representation in future studies.

5. The example data in Fig. 1 provide little additional perspectives on the “main result” of (5). Panel C shows diagonal elements of the T matrix. Is it correct that this is a reciprocal system? If so, why are both real and imaginary parts shown when the axis label reads $\text{Im } T$?

We return to the “additional perspective” in our response to the next question. As for panel C: There are two vertical axes: a left-hand side for $\text{Im } T$, and a right-hand side for $\text{Re } T$. We have added arrows to the two axes to try to make them more obviously visible.

6. The fact that some of the traces exhibit consecutive oscillator-like responses is not surprising, given that the quantities being plotted are analogous to the circuit admittances seen by the incident field at each location.

We agree that an admittance viewpoint is a helpful additional mechanism for understanding the lineshapes in Fig. 1(c), and we now include repeated mentions of admittance matrices in our Results section for additional explanation.

However, a volume-admittance-matrix viewpoint of full-wave scattering problems (as opposed to quasistatic or nearly quasistatic scenarios) is often not useful! The distributed elements typically offer no discernible explanatory power and the quantities are hard to compute (they are matrix inverses, as described in Sec. I of the SM). This is likely why classic books such as Newton’s *Scattering Theory of Waves and Particles* or Nussenzveig’s *Causality and Dispersion Relations* make no mention of impedance or admittance matrices.

In this sense, our work shows *why* a volume-admittance-matrix formulation of causal scattering theory can be useful. Not for modeling (where the computational burden is too much), but for identifying fundamental limits, where the totality of the mathematical structure implied by passivity is uniquely well-suited. Our goal in Fig. 1 is to show the *combined* structure of the volume admittance matrix.

We think some readers may have similar questions about connections to impedance and admittance matrices. One key point is the following: it is *not* possible to just use any impedance or admittance matrix and end up with a useful formulation. Consider a finite-sized scattering body. There are at least six relevant matrices that can be formed:

- A scattering matrix, defined through channels at a bounding surface (this is arguably the most common matrix approach in electromagnetic theory; certainly it is the most common in nanophotonics)
- An impedance matrix at a bounding surface
- An admittance matrix at a bounding surface (this is Waterman’s T matrix)
- A scattering matrix in the volume. (We have never seen one defined, but it could be done via the abstract transforms discussed in Sec. IIIA of the SM.)
- An impedance matrix in the volume.
- An admittance matrix in the volume (the T matrix that is the subject of this manuscript)

Of these six, only the volume admittance matrix appears to provide a useful formulation for spectral bounds. For the surface quantities, causality has tricky subtleties involved. The “channels” at the surface are orthogonal basis functions that are spatially distributed, which makes it non-trivial to identify the time at which scattering response occurs relative to the origin of time of an incoming wave. (This issue is discussed extensively in Nussenzveig’s *Causality and Dispersion Relations*.) The result is that KK relations have an extra size- and shape-dependent phase shift in the real and imaginary parts of the matrix elements. This appears to inhibit any hope of combining positive passivity constraints with a KK relation. (There is a second issue: surface-matrix sum rules are quite hard to identify, and a third one: any such sum rule almost surely would not satisfy domain monotonicity.) For the volume matrices, the high-frequency response of the impedance matrix is difficult to generically identify, without which sum rules cannot be derived. And the volume scattering matrix would appear to suffer from the same issue. Hence, the volume admittance matrix truly is unique in how it seamlessly combines KK relations, passivity, and sum rules. We believe this is useful to demonstrate with a figure.

Throughout the main text, we have removed the word “surprising” in regards to Fig. 1. We feel that many readers from the scattering-theory community may find the combined structure shown in Fig. 1 to be surprising, but we leave that for them to decide.

7. If the bottom panel were taken in isolation, it could be interpreted as clearly defined cascade of oscillator circuits. The central panel taken in isolation would not lead to that conclusion. The caption text stating that this is a “new type of matrix-valued, nonlocal scattering oscillator” overlooks the prevalence and well-studied nature of these responses in systems as simple as a lossy Fabry-Perot etalon.

Here we believe the reviewer is conflating the oscillators of our representation (which we refer to as “mathematical oscillators”) with the “physical” oscillators used for systems such as Fabry-Perot etalons. The oscillators of our representation do not correspond to poles in the lower-half plane, which is crucial: the only degrees of freedom are the matrix coefficients, whose constraints are bounded, convex sets. (In other words, the constraints prevent blow-up: they are “bounded below” by positive definiteness requirements, and “bounded above” by the sum rules.) By contrast, the resonances (scattering-matrix poles) of Fabry-Perot etalons are quite useful for modeling and understanding (interpreting line widths, free-spectral range, etc.), but they would not be useful from a fundamental-limits perspective. How many resonances is one allowed to have in a free-form shape? Where can the poles lie in the lower-half plane? We cannot put useful constraints on these quantities just from fundamental considerations.

A useful test for whether similar ideas to ours have been put forward in the Fabry-Perot literature is to search for a single paper in which Fabry-Perot “oscillators” are discussed in tandem with sum rules and passivity constraints; we know of no such papers.

8. Panel D shows the matrices T_i at three frequencies, which I assume are taken directly from the computed T matrix at the selected frequencies via (4). In my opinion, these pictures and their supporting text provide little quantitative or qualitative information to the reader. What data or patterns in those pictures “showcase the extent to which passivity and causality restrict the possible form of the scattering T matrix”?

We agree that the previous panel D was not clearly helpful. We have removed it. We have also removed the previous panel F (the susceptibility plot), which we felt also was not clearly helpful.

9. In Panel E, the positivity of the eigenvalues of the $\text{Im } T$ matrix is demonstrated. The PSD-nature of this matrix (which represents many things, such as the admittance parameters of a circuit network) is well-known throughout electromagnetics, from optics to microwave circuits. Its applications range from the verification of measurement data to constraining optimization problems. By showing that $\text{Im } T$ has positive eigenvalues, this panel serves only to verify that the numerical modeling accurately enforces the system's passivity.

In summary, Fig. 1 does not demonstrate any new information. The statement that it helps in the paper "uncovering this structure [of the T matrix]" (referring to the results near (5)) is a very big stretch.

We have made two content changes: first, in panel E, we now have visualizations of the sum rule convergence for three different scattering bodies. Moreover, we include one scatterer that occupies the full elliptical domain, and two scatterers that occupy smaller domains within the ellipse. This allows us to include in panel E a demonstration of a crucial subtlety of the sum rules: that they are *domain monotonic*, which means that the sum rule for any sub-domain will necessarily be no greater than (in a positive semidefinite sense) the sum rule for the full designable domain itself. Without this, it would be impossible to formulate bounds that apply to all possible scatterers within a region.

Hence, we feel that Fig. 1 now appropriately encapsulates the key idea of our formulation: a single matrix, the T matrix (the volume admittance matrix), simultaneously combines KK relations, passivity, and sum rules in a way that is uniquely useful for identifying bounds.

10. The example calculation demonstrating the limits on NFRHT is very interesting, and I appreciate the details given in the SM. However, the manuscript's single page and figure devoted to this complex derivation, its results, and subtle interpretations doesn't do the work justice. The 10 pages of SM on this topic contain a considerable amount of technical work that should be the subject of its own critical review.

We have revamped our paper to match the *Nature* formatting style, which allows us to bring key derivations, such as the of NFRHT, into the Methods section of the main text. The editor also granted us an additional 1500 words in the maximum word count for the main text, which we used to add a second figure (the "sharp-tip" figure) related to NFRHT, alongside a corresponding discussion. (We also felt the "domain monotonicity" derivation is quite important and brought that into the Methods section as well.) The new Methods section is not overly long – our previous SM was double-spaced and had a lot of redundant and extraneous information – but we believe the current manuscript is certainly stronger for having these discussions and derivations in the main text, and we thank the reviewer for the comment.

11. The final section of the paper is an excellent forward-thinking discussion of open problems in electromagnetics, acoustics, and quantum physics. I agree that the study and application of the T matrix (or its many other names from various integral equations, SM

Sec. II) will likely be a fruitful path forward in many areas.

We thank the reviewer for the positive appraisal.

12. In conclusion, the 32 pages of scientific content of the SM is very interesting, however I believe that the paper itself is an overly condensed abstract of this very broad work. The example data shown in Fig. 1 do little to support the “uncovering this structure [of the T matrix]” and the NFRHT summary is, realistically, far too dense to comprehend without a detailed study of 10 pages of SM.

I would strongly suggest that the authors consider dividing this work up into more traditional journal articles on their work regarding the structure of the T matrix for non-reciprocal systems (SM Secs. I-VI) and the application of those results to NFRHT (SM Secs. VIII-IX). This would make the technical content of the work more accessible and would allow for more focused scientific reviews.

We thank the reviewer for their thoughtful comments on the structure of our manuscript. We have made three significant changes to the content:

1. We have a more compact and complete derivation of the key representation (Eq. 6) in the main text of our manuscript. This cut down significantly on the amount of information that we needed in the SM (some of which was admittedly redundant), and in fact is more general, as we now have the nonreciprocal components included in the representation.
2. We moved the two key derivations (domain monotonicity, NFRHT heat-transfer bounds) to the Methods section of the main text in a Methods section.
3. We include the sharp-tip NFRHT discussion and the corresponding figure in the main text.

With these three changes, the main text now can essentially stand alone; the SM exclusively consists of implementation details, relevant background, and contextual discussions for readers interested in very particular points.

Reviewer 2

1. The work by Zhang, Monticone and Miller reports on a new approach to derive fundamental bounds on the performance of scattering nanophotonic structures. The approach is based on the new realization that the “T-matrix” of a scatterer can be decomposed as a sum of T-matrices that each have the frequency dependence of a Lorentz oscillator, and bounded by causality and passivity.

In my view, the paper is very interesting, and the application to the near-field radiative heat transfer problem shows that the oscillator decomposition indeed allows to derive fundamental bounds.

We thank the reviewer for the positive comments.

2. At the same time, I believe that the paper may not be suited for Nature Communications, at least in its present form, for the following reasons:

All the “meat” of the paper is actually in the supplement. Thus the paper is in fact more a

manual to the actual description of the theory, and in itself not very clear. This means the paper does not really do justice to its content.

We thank the reviewer for this comment, which is quite similar to the final comment of Reviewer 1. We agree that the previous manuscript did not adequately stand “on its own” without the Supplementary Materials. We have made three significant changes to the content of the manuscript (copying these points from above):

1. We have a more compact and complete derivation of the key representation (Eq. 6) in the main text of our manuscript. This cut down significantly on the amount of information that we needed in the SM (some of which was admittedly redundant), and in fact is more general, as we now have the nonreciprocal components included in the representation.
2. We include the two key derivations (domain monotonicity, NFRHT heat-transfer bounds) in the main text in a Methods section, as appears to be common in *Nature*-style articles.
3. We include the sharp-tip NFRHT discussion and the corresponding figure in the main text. (The editor gave us permission to add some length to the main text.)

(We will also note here that we rewrote the introduction according to the strict *Nature* style; the space we saved by using this approach was also helpful in enabling more technical content in the rest of the paper.)

The main text is now 10 pages long (without references), and essentially can stand alone. The SM is now 7.5 pages long (excluding references and the title page), and primarily contains background review material (Sec. I, Sec. IIIA), simple algebraic derivations (e.g. the T-matrix symmetries), and implementation/calculation data (Sec. VI). We believe the “meat” of the paper is now almost entirely concentrated in the main text. We thank the reviewer for the comment, as we believe the paper is better for these revisions.

3. To illustrate how this paper is “not very clear”: For instance, surrounding Eq. (4) and (5) and Figure 1, the authors state that “a discrete set of frequencies ω_i ” is chosen but they never state how this choice is made, given a problem. From the supplement it appears that the ω_i are chosen simply as an equidistant discretization of the frequency axis, but instead when discussing Fig 1c on page 3 (second column), the reader is supposed to recognize oscillators. Indeed, one sees of order 8 resonances in the spectrum. However, as these actually have a distinct frequency width, the relation between these resonances, how these oscillators could be lossless, and how they differ from the signatures of the QNMs from literature, is unclear.

We thank the reviewer for this thoughtful comment. In our revised version of the manuscript, we do away with the notion of discrete frequencies, and just use a continuous-frequency representation. (Now the T matrix is an integral over matrix coefficients, instead of a sum.) So just as we think of electric fields as continuous functions of frequency, and can compute them at any frequency, the same is true of the T-matrix components.

The second question from the reviewer is that, for example, in part (c) one sees about 8 resonances in the spectrum, but these all have nonzero widths. How do we reconcile this with our claim of lossless oscillators?

The key idea is that the underlying (“physical”) resonance structure of the problem dictates aspects of the lossless-oscillator structure as well. If we look at the lineshape for $\text{Im } T$ for any of the panels in Fig. 1(c), we can imagine *every frequency* as corresponding to a unique, lossless oscillator. In the third panel of Fig. 1(c), for example, the $\text{Im } T$ lineshape comprises sequential Lorentzian-like lineshapes. If we consider the peak of one of those lineshapes, that represents one lossless oscillator. If one perturbs the frequency by a small amount, we are now considering a new lossless oscillator. But the amplitude of this oscillator cannot be too different from the peak-oscillator amplitude, as the underlying resonances of the system have a limited Q factor. This leads to a restriction on how rapidly the oscillator coefficients vary as a function of frequency.

Hence, every frequency has its own oscillator. However, if one has a continuous distribution of positive imaginary parts, the real parts will simply be linear combinations of the real parts of those lossless oscillators, which together “look like” a lossy oscillator. Of course, this must be the case: one could represent this system with quasinormal modes and lossy resonances, and indeed, such a model would be better, in the sense that it could fit the data with many fewer parameters. However, the lossless-oscillator representation is better from an optimization perspective; the positivity traits seen in Fig. 1(d) and 1(e) of the revised manuscript would not be possible with the coefficients of a lossy-oscillator model.

In our discussion of Fig. 1 in the last paragraph before the “Ultimate limits to NFRHT” section, we now clarify:

The lineshapes of the T-matrix elements closely mimic the Drude–Lorentz-like behavior of electronic transitions, but they arise not from real material oscillators, but from complex wave-scattering behavior itself. Collectively, there are nonzero lineshape widths thanks to the underlying resonant physics, but every frequency can and should (for our purposes) be interpreted as having its own, lossless-oscillator amplitude, given by $\omega \text{Im } T(\omega)$.

4. I believe that for this paper to resonate with the reader, it should be made much clearer how the authors construct the oscillators ω_i for the given scattering problem. Supposing that they have access to the calculated full T-matrix versus frequency, then how would they go about defining the oscillators ω_i , and using the resulting construction to their advantage?

Now that we do not use any discretization, and we define things continuously, hopefully the ambiguity regarding the discrete oscillators is gone. One can simply compute $T(\omega)$ at any frequency ω , and the quantity $\omega \text{Im } T(\omega)$. In a reciprocal system, this quantity is exactly the matrix oscillator amplitude. In a nonreciprocal system, the complex-symmetric part of $\omega \text{Im } T(\omega)$ is the matrix amplitude $X(\omega)$, while the skew-symmetric part of $\omega \text{Im } T(\omega)$ is the matrix amplitude $Y(\omega)$. (Both X and Y appear in our revised formulation, see Eq (6) of the revised manuscript.) Hence computing the oscillator amplitudes is straightforward. (Or, at least as straightforward at computing $T(\omega)$, which does involve computing full inverses...) We thank the reviewer for this comment, which in part inspired the switch to continuous frequency, as we think the results are stronger in their new form.

As for how we use the resulting construction to our advantage, we have more discussion throughout the Results section of how the mathematical structure of the formulation is advantageous. At the end of paragraph two of the results section we point out:

While Eq. (3) reduces the degrees of freedom to the anti-Hermitian part of T , additional passivity considerations are needed to meaningfully constrain the possible scattering response.

Then at the end of paragraph 3 we continue this thread:

Using the symmetry relations for $X(\omega)$ and $Y(\omega)$ around $\omega = 0$, we have the constraints $X(\omega)+Y(\omega) \geq 0$ and $X(\omega)-Y(\omega) \geq 0$ at positive frequencies, which further imply $X(\omega) \geq 0$. These constraints are convex (though still unbounded) in $X(\omega)$ and $Y(\omega)$.

And, finally, after Eq (6):

The collective representation of Eq. (6) is the foundational result of our paper: the T matrix of any linear scattering body must be decomposable into a set of lossless oscillators, with matrix-valued coefficients satisfying definiteness conditions and constrained in total strength. The only degrees of freedom in the scattering process are the matrices $X(\omega_i)$ and $Y(\omega_i)$, both of which have strong constraints on the bandwidth over which they can be nonzero. The $T(\omega)$ matrix is linear in these matrix degrees of freedom and the constraints are bounded convex sets.

We also believe the NFRHT result is a compelling example of how we use this construction advantageously.

5. The authors claim a distinction with the field of QNMs, but at least in the QNM field there are recipes for how to find them in the first place, and how to use QNMs to calculate observables. For the example of figure 1, to highlight the claimed distinction with QNMs, it would be helpful to highlight both the real-valued oscillators at ω_i quantitatively, and any QNMs that readers may wish to read into Fig. 1c instead.

Without such pointers I believe it will be impossible for an interested reader to take up this result, and apply it to a problem of their own choosing. How to apply the method to derive a bound for a problem of choice other than the example, or if the method could even be used for design, is not clear from the paper.

We agree that this is an important point. We now make clear that the lossless oscillators should be interpreted as *individual oscillators at every frequency*. Much like a material susceptibility that satisfies KK relations can be interpreted as having lossless oscillators at every frequency (as is often derived *ab initio* in quantum susceptibility models). As discussed above, there is a simple recipe to calculate the oscillator amplitudes via $\omega \text{Im } T$.

We considered the following additional SM figure: a plot that shows the pole frequencies of computed QNMs, in the lower-half of the complex-frequency plane, alongside the continuum

of real frequencies for the lossless oscillators (which would just be a solid line from 0 frequency to infinity). The plot would look like:

However, we felt that such a figure would not convey any particularly helpful information. Instead, we believe the revisions discussed above (the continuous frequency representation, the discussion of the oscillator amplitudes given by $\omega \operatorname{Im} T(\omega)$, etc.) should clarify this point.

As for how to use this representation, we have added as the third sentence in the Discussion section:

We propose a recipe for identifying fundamental limits: rewrite any objective of interest in terms of the $T(\omega)$ matrix, and then use the representation of Eq. (6) as the constraints.

As we discuss throughout the manuscript, the constraints are a bounded, convex set, which makes them amenable to standard optimization methods (e.g. cf. *Convex Optimization* by Vandenberghe and Boyd).

We also discuss in the second paragraph of the Results section how additional single-frequency constraints could be seamlessly merged with this representation.

6. The fact that the oscillators are real-valued is claimed as crucial by the authors. Yet the authors introduce loss as $i \omega \gamma$, (Eq. 5), and do not discuss how to deal with the problems that surely must arise in applying the limiting procedure $\gamma \rightarrow 0$. Here I note that also the supplement on this point is surprisingly brief – so much so that the fact that a limit must be taken is not even written down let alone discussed for the NFRHT example.

We thank the reviewer for this comment. The lossless nature of the oscillator is crucial *not* because of how the loss is taken to zero. In fact, for any objective that has any nonzero bandwidth (which is any problem for which a spectral representation might be useful), the contributions of the lossless oscillators are integrated over frequency, and the details of the implementation of the loss rate matter little, if at all. This is why it did not need to be discussed in the NFRHT case, for example. In our revised manuscript, we now actually never discretize the frequency domain in our NFRHT bound derivation, and we just directly apply the sum-rule and positive semidefiniteness constraints. So hopefully this question will not even arise for readers, given these changes.

The key feature of the lossless nature of the oscillator is that it *removes* extraneous parameters in which any optimal design problem becomes nonlinear and nonconvex. (As discussed in Sec. IV of the SM, the pole frequencies are problematic parameters in QNM models, from an optimization / fundamental-limit perspective. So are the coupling constants,

as discussed there in more depth.) We have tried to clarify the discussion of the mathematical structure of the formulation and its connections to convexity and bounded throughout the Results section of the manuscript, particularly in the three examples that we gave in response to Question 4 above.

7. Finally I note that the notion of T-matrix presupposes that one makes a separation between scatterer, and background of which one must know the Green function. I find that the paper is unclear about how this distinction must generally be made, and for instance how one deals with resultant problematic properties of G (e.g., guided modes in planar systems), or in the scatterer (e.g., Equation 17 of the supplement assumes that the scatterer has a finite extent – but it appears that the NFRHT is infinitely spatially extended).

We thank the reviewer for raising this very important point. Issues with G are not directly a problem; one can always define a suitable background/scatterer separation, there is a well-defined, unique T matrix for such a separation, and we want to search for a meaningful representation. The problem is how to formulate the causality and passivity conditions (in some cases these would be indirect symptoms of problematic properties of G), which arise for *any* approach to spectral response that uses KK relations or sum rules. Now, when we introduce the T matrix, we mention the assumption of a vacuum background in order to use causality principles (two sentences after Eq. 1):

A passive scatterer in vacuum has a causal response function...

Where these issues arise in fact dovetails nicely with our discussion of historical issues with developing sum rules in acoustics. For example, a homogeneous background with a vacuum scatterer appears to violate causality in the scattered fields, for exactly the same reasons as the acoustic-scattering anomalies that prevent sum rules. And, in some cases, the T-matrix approach may alleviate these issues. We now include this as the final sentence of the paragraph about classical wave scattering and acoustics, at the bottom of page 6 / top of page 7:

(Relatedly, wave scattering with any non-trivial/non-vacuum background has historically stymied sum rules, and this is another avenue of exploration with the T matrix.)

Regarding the infinite extent of the scatterer in the NFRHT case: the scatterers are responding to an infinite line of point sources (after using reciprocity to move the sources to the separating plane), and we bound the response to each source individually. We can regard each scatterer as large but finite-sized to compute this bound, then take the limit of infinite size. The fields decay quickly so that there is no issue with this limit. Hence the sum rules and positive semidefinite constraints are well-behaved. We now include a discussion of this limit in the NFRHT derivation in the Methods section of the main text:

Equation (20) represents the total flux from an infinite plane of sources between the infinite bodies. An upper bound on this flux is given by the upper bound on the flux generated by a single set of point sources at a given position on the separating plane, multiplied by the (infinite) area of the plane. This allows us to easily switch to the quantity of interest in large-area NFRHT: the per- area radiative heat transfer, which is bounded above by the maximum flux from a

single set of sources at a single position on the separating plane. This also resolves a second possible difficulty: how to represent the T matrix for infinite, extended structures? For point sources in the near field, there is no issue: the fields decay sufficiently quickly that the response is guaranteed to be well-behaved. (Intuitively, one can imagine substituting large but finite-sized structures at this stage, and later taking the limit as size goes to infinity. The rapid field decay ensures that the subsequent integrals converge, even in the infinite-size limit.

Reviewer 3

This paper presents an interesting theoretical analysis of electromagnetic scattering. It proposes a new mathematical representation of the T-matrix that incorporates the principles of causality and passivity. The authors use this representation to uncover hidden resonant patterns in scattered fields. As an application, the authors develop a general theory of maximum radiative heat transfer in the near field that provides a better theoretical bound.

I enjoyed reading this paper as it contains many novel thoughts. It provides a different viewpoint on wave scattering from the coupled-mode theory or quasi-normal mode expansion. This new viewpoint is particularly useful in treating physical problems involving absorption and emission.

We thank the reviewer for the careful reading and the positive appraisal.

The main results of this paper can be re-derived as follows:

The T-matrix is causal and thus satisfies the Kramers-Kronig relation:

$$\text{Re } T(\omega) = 1/(\pi\omega) * \text{Im } T(\omega) \quad (\text{a})$$

where * denotes convolution. Therefore,

$$T(\omega) = \text{Re } T(\omega) + i \text{Im } T(\omega)$$

$$= \int_{-\infty}^{+\infty} [1/\pi(\omega'-\omega) + i\delta(\omega'-\omega)] \text{Im } T(\omega') d \omega'$$

$$= \lim_{\epsilon \rightarrow 0} \int_{-\infty}^{+\infty} 1/\pi(\omega'-\omega - i\epsilon) \text{Im } T(\omega') d \omega' \quad (\text{b})$$

For a reciprocal system, $\text{Im } T(-\omega') = -\text{Im } T(\omega')$, then Eq.(b) becomes:

$$T(\omega) = \lim_{\gamma \rightarrow 0} \int_0^{+\infty} 2\omega'/\pi(\omega'^2 - \omega^2 - i\gamma\omega) \text{Im } T(\omega') d \omega' \quad (\text{c})$$

where we define $\gamma=2\epsilon$. This is Eq.(5) in the paper once one substitutes Eq.(4) into Eq.(c). If the system is passive, then $\text{Im } T(\omega')$ is positive semidefinite.

We thank the reviewer for this very compact derivation. In our rewritten manuscript, we have altered our derivation, very much inspired by this one. (There were a few modifications necessary to include the nonreciprocal part, including starting with ωT instead of just T , but otherwise the approach is similar.) Thank you!

Now the mathematical content is clear. Causality reduces the degree of freedom for $T(\omega)$ by half: Knowing $\text{Im } T(\omega)$ is sufficient to reconstruct $T(\omega)$. Moreover, the integral representation of $T(\omega)$ has a Lorentzian-like weight factor. These observations lead to the title of this paper: All electromagnetic scattering bodies are matrix-valued oscillators.

Eq.(c) is a new mathematical representation of the T-matrix that incorporates causality and passivity. Whether it is useful to physicists depends on, of course, what new physical

insights this mathematical representation can provide. Revealing these physical implications is the main contribution of this paper. The authors have successfully illustrated how this mathematical representation can reveal hidden patterns in wave scattering using concrete physical examples (Fig.1). They also demonstrate how this representation can shed light on an open question on the theoretical limit of near-field radiative heat transfer (Fig.2). The new theoretical bound is much closer to the state-of-the-art performance.

As such, I think this work is a solid and important theoretical work that merits its publication in Nature Communications. It should be of interest to a broad audience in the physics community.

We thank the reviewer for the careful reading and positive recommendation.

Nonetheless, I do have a few comments for the authors to improve their manuscript.

1. The authors have focused on the T -matrix. Another fundamental object in wave scattering is the S -matrix. I was wondering whether the authors' results on the T -matrix have any implications on the S -matrix. A discussion on this issue would be useful.

We agree that this is quite an important question. We now include a discussion of this in the new third paragraph of the Discussion section:

One might wonder why we have utilized the $T(\omega)$ matrix, when the vast majority of photonics theory uses the scattering matrix $S(\omega)$? There are two reasons. First, in many scattering systems, incoming and outgoing waves are spatially distributed, requiring exquisite care with the causality conditions, leading to (for example) phase shifts in the S -matrix KK relations [13]. It becomes unclear which degrees of freedom are necessary, sufficient, and have convex passivity constraints. The second issue is that there is not, as far as we know, a useful S -matrix sum rule of a positive semidefinite quantity. Without such a sum rule, all response is unbounded. As discussed above, volume T matrices appear to be the unique scattering/impedance/admittance matrix where KK relations, passivity, and sum rules can all be combined into a bounded, convex set of constraints.

There is a related, broader question (initiated from another reviewer comment), which is that KK relations arise for any impedance/admittance/scattering matrix. Why did we choose the volume T (admittance) matrix? We now discuss this question at the end of paragraph 2 of the Results/Passivity Constraints section:

For any scattering problem there are at least six matrices with a form similar to Eq. (3): a scattering matrix, an impedance matrix, and an admittance matrix, each defined either in the volume or at a bounding surface. Yet only one of those six—the volume admittance matrix (essentially, $T(\omega)$)— appears to be useful for bounds. While Eq. (3) reduced the degrees of freedom to the anti-Hermitian part of T , that is not sufficient to ensure finite response: additional passivity and sum-rule considerations are needed to meaningfully constrain the possible scattering response.

2. The meaning of Eq.(4) needs to be clarified. Strictly speaking, the left is a matrix-valued function, while the right is a distribution. It is not clear to me how equality can be established here. In addition, there is a typo in Eq. (4): it should be $\delta(\omega-\omega_0)$.

That equation meant equality in a distributional sense (averaged over any nonzero bandwidth); however, we now just use an integral equation (along the lines of the reviewer's derivation) and there is no need for such a summation. (Nor for the delta function.)

3. The authors have written the conditions $\sum_i T_i = I$ and $\sum_i T_i \leq I$ simultaneously. It seems that Eq.(3) implies $\sum_i T_i = I$. Where does the weaker condition $\sum_i T_i \leq I$ come from? Why do we need it?

We thank the reviewer for catching that we have not clarified our transition from the equality to the inequality. In fact, the inequality is the *stronger* condition, in the following sense. For any given scatterer volume V , the sum rule for the T matrix leads to the integral equality, where the right-hand side is proportional to \mathbb{I}_V , which is the identity matrix on that domain. However, in a design problem, we do not know the exact scatterer domain a priori. Instead, we have a designable region D that contains all possible designs / domains V . The monotonicity property of our sum rule implies that the sum-rule constant matrix of *any domain within the designable region* will be bounded above (in the positive definite sense) by the sum-rule constant matrix for the designable region. Hence, the inequality constraint involves the identity matrix as defined on the entire designable region, and allows us to consider all possible designs within a designable region.

We have added a new paragraph immediately after Eq (5) to clarify this key point:

For design problems, one considers many possible scatterer domains V , each of which has different matrices on the right-hand sides of the sum rules of Eqs. (4,5). How, then, can one accommodate many possible designs? Here we can again make the (critical) choice of the Hermitian/anti-Hermitian split in the KK relation, which, as we prove in Methods, endows the sum rules with a monotonicity property: enlarging V can only increase (in a positive semidefinite sense) the right-hand sides of Eqs. (4,5). Hence, for a designable domain D containing all possible scatterer sub-domains, we can convert the equalities of Eqs. (4,5) for specific volumes V into inequalities over the designable domain D .

4. Figs. 1c and 1d: The authors have only presented the results on the diagonal elements of the T -matrix. What about the off-diagonal elements of the T -matrix? Do they exhibit similar behaviors?

We thank the reviewer for this suggestion. We now include the off-diagonal elements. They show some similarities to the diagonal elements (they do satisfy similar entrywise KK relations), but now they do not have a positivity constraint on their imaginary parts. The only positivity conditions are positive semidefiniteness of any diagonal sub-block containing them, which we partially represent in Fig. 1(d), which shows all of the eigenvalues being positive. Hence the off-diagonal elements of the T matrix seem "less orderly" than their diagonal

counterparts.

Overall, I think the authors have done an excellent job of presenting their new mathematical representation of the T-matrix and demonstrating its usefulness in understanding wave scattering. I believe that addressing the above comments will further improve the manuscript.

We thank the reviewer again for the positive review, as well as the helpful comments. We believe that addressing these comments did indeed improve the manuscript.

REVIEWER COMMENTS

Reviewer #1 (Remarks to the Author):

The authors have addressed most my concerns and I appreciate the extensive changes to the main manuscript and supplemental materials, as opposed to simple edits. The added emphasis on the connections between the T matrix and the admittance matrix of passive systems clears up many aspects of the work. Overall, the paper is now much more readable on its own. I'm hopeful that the approach presented here will have an impact on the broader scattering and electromagnetics communities.

Remaining small concerns (mainly encountered while replicating results in Fig. 1):

1. The example calculation specifies a nondispersive susceptibility that does not conform to the assumptions above (4).

2. Related to the previous point, is the bottom row of Fig. 1e obtained from the computational model or are the matrices computed analytically as scaled identity matrices? The text states that “the sum rules converge to the low-frequency matrix constant”, but the upper limit of integration for the final “infinite case” is not listed.

3. Caption of Fig. 1 “scalar multiple of the identify” is likely a typo.

4. I still seriously question what the authors mean by “oscillator-like response consistent with the KK relation” in describing the curves in Fig. 1c. The self-terms resemble the admittances of oscillator circuits, but the cross-terms appear to be somewhat random curves roughly centered around zero, much like the curves in Fig 1b which the authors themselves call “seemingly random”. What kinds of curves would the authors not consider as “oscillator-like responses”? As the authors state in their answers to previous questions, the elements of the volume impedance matrix “typically offer no discernable explanatory power”, yet it is these elements that are plotted in the figure and stated to clearly demonstrate a specific type of behavior.

5. The supplemental material Sec. VI states that five sample locations are selected, but only three are listed and used.

Reviewer #2 (Remarks to the Author):

The review comments of round 1, including my own, have consistently pointed out that this was a very interesting and impactful work at heart, but with presentation issues (whereby the appreciation was for the supplement more than the paper). The authors have very strongly revised the paper, and in my view this has led to a spectacular improvement: the power, distinguishing factors, and beauty of the original claims and work now really shine in the main manuscript, and some confusions that were originally introduced by the presentation are now completely avoided.

In my view, this paper should now certainly be published perhaps even as-is in Nature Comm. If I may point at one minor shortcoming in presentation that I consider it would be a good idea to fix, then it is that for Figure 2c and d it is nowhere specified what the "optimal 2D heterostructure" and "optimal Drude" structures entail. A specification in the supplement of household details (e.g., for the $T=300$ K case, what are these quoted optimal structures, perhaps how they were found/how the calculation was done) would solve this. The caption should explain also what the asterisks in Fig. 2c specify.

Reviewer #3 (Remarks to the Author):

The authors have effectively addressed all my comments and made significant modifications to strengthen the manuscripts. As a result, I can recommend the publication of this paper in Nature Communications.

Referees 1 and 2 have raised similar concerns regarding the original manuscript, noting that many interesting results were only included in the Supplement. I agree to some extent with Referees 1 and 2, as it would have been preferable for the authors to include these discussions in the main text. However, it is to the credit of the authors that they have rectified these deficiencies by rearranging the paragraphs and extensively revising the main

text.

I am also pleased that the authors have found my alternative derivation helpful in streamlining their presentation.

Reviewer: Dr. Cheng Guo

All reviewer comments are included below in black text; our responses are in blue text. We thank all reviewers again for their time and their thoughtful comments. We are glad to see that the reviewers appreciated the modifications that we made to the manuscript, and all appear to agree on the potential impact of our work.

First we summarize the contents of this response document: Reviewer 1 raised a few interesting questions, which we have fully addressed, as well as suggestions (e.g. typos) that we have corrected. In particular, we have rigorously clarified what we mean by an “oscillator-like response,” and we have made a minor change to the material model considered in our example, which is now weakly dispersive and fully consistent with the necessary asymptotic response. None of the figure data is perceptibly different. Reviewer 2 had a helpful suggestion about a small amount of data and text clarification to add in the main text and Supplementary Materials, which we have done. Reviewer 3 did not suggest any further revisions.

Reviewer #1:

1. The example calculation specifies a nondispersive susceptibility that does not conform to the assumptions above (4).

We thank the reviewer for this comment. In that example, we wanted to clearly distinguish between “material oscillators” (transitions in the material permittivity) and “scattering oscillators” (transitions in the scattering response of the system). We chose a dispersionless material so that there could be no confusion that we might simply be “repackaging” the underlying material response. A dispersionless material is indeed inconsistent with our assumptions, but a very minor change resolves this: simply choose oscillator and plasma frequencies for the material well above the simulated frequencies of interest. We use dimensionless numbers for that example, such that oscillator and plasma frequencies above 10 ($2\pi c/a$) (with their ratio keeping the low-frequency permittivity fixed) is sufficiently large for demonstration purposes. (Note that such conditions are not uncommon in practice even at optical frequencies: small-dispersion dielectric materials at infrared frequencies of a few hundred milli-electron volts can have plasma frequencies on the order of 15 eV.) That example is one in which the low-frequency sum rule is the binding one, not its high-frequency counterpart, so from a bound/response perspective it does not matter what values we choose for the oscillator and plasma frequencies, beyond their ratio.

We agree that it is important to clarify this point in the manuscript. We have made the following change: we now choose a plasma frequency of 20 ($2\pi c/a$) for our example; this introduces negligible dispersion and negligible change in the actual scattering response at the relevant frequencies (verified by full-wave simulations). In our description, we now write:

to clarify the origin of the oscillators, we use a material with $\chi = \frac{w_p^2}{w_0^2 - w^2 - 1i g w}$, with $w_p = 20$, $w_0 = 10$, $g = 0.01 w_p$, which is nearly dispersionless with $\chi = 4$ for w between 0 and 1 (all frequencies in unit of $2\pi c/a$) and consistent with the necessary asymptotic response.

2. Related to the previous point, is the bottom row of Fig. 1e obtained from the computational model or are the matrices computed analytically as scaled identity matrices? The text states that “the sum rules converge to the low-frequency matrix constant”, but the upper limit of integration for the final “infinite case” is not listed.

Thank you for the careful reading. They are obtained from the computational model, and the final “infinite” case is really the convergent numerical approximation of the limit of the integral as its upper limit goes to infinity. The integral is done with a 2000-point Gaussian quadrature, and the error is measured by Frobenius norms of the matrices. We now clarify in the text:

(the numerical integral converges to <1.7% error, as measured by the matrix Frobenius norm, using a 2000-point Gauss-Legendre quadrature from frequency 0 to 40 ($2\pi c/a$))

3. Caption of Fig. 1 “scalar multiple of the identify” is likely a typo.

Indeed it is, thank you. We have corrected “identify” to “identity”

4. I still seriously question what the authors mean by “oscillator-like response consistent with the KK relation” in describing the curves in Fig. 1c. The self-terms resemble the admittances of oscillator circuits, but the cross-terms appear to be somewhat random curves roughly centered around zero, much like the curves in Fig 1b which the authors themselves call “seemingly random”. What kinds of curves would the authors not consider as “oscillator-like responses”? As the authors state in their answers to previous questions, the elements of the volume impedance matrix “typically offer no discernable explanatory power”, yet it is these elements that are plotted in the figure and stated to clearly demonstrate a specific type of behavior.

When we use the term oscillator, we are referring to functions of the form

$$\frac{1}{\omega_0^2 - \omega^2 - i\gamma\omega}$$

where γ is positive. Any function not of this form (or decomposable into this form) is not an oscillator. Examples can be generated easily (e.g., $e(-a|\omega|^2)$ or $(1+i)e^{-a|\omega|^2}$), but more generally, what are the characteristics of functions composed from Drude-Lorentz forms? Their real parts tend to increase from minima to maxima, undergo transitions back to-minima, and repeat this process. Their imaginary parts are positive, and the widths of the peaks of the imaginary parts are correlated with the widths of the transitions in the real parts. Of course these statements apply generally to causal response functions described by Hilbert transforms, but that is the origin of our work: identifying specific response functions described by Hilbert transforms, which by linear-response theory can be represented as linear combinations of oscillators.

What are examples of functions in electromagnetic scattering not described by oscillators / Hilbert transforms? Take a continuous-wave incident field such as

$$E_{\text{inc}}(\omega) = e^{-i\omega_0 t} \delta(\omega - \omega_0).$$

This function is oscillatory, but it is *not* an oscillator. This is because it is not itself a causal response: in the time domain, the field (ignoring constant pre-factors) is $E_{\text{inc}}(t) = \text{Re}[e^{-ik_0(x-ct)}] = \cos(k_0(x-ct))$, where $k_0 = \omega/c$. This field exists at all times, so its

frequency-domain components are not related by a Hilbert transform. And the shape of its real and imaginary parts are nothing like that which we described in the paragraph above.

Another example of a function whose real and imaginary parts are not related by Hilbert transforms is the scattered field in Fig. 1(b), raised by the reviewer in their comment. To understand this more concretely, consider the fields scattered by a small dipolar scatterer with polarizability $\alpha(\omega)$. The scattered field at x generated by a scatterer at x' is given by (using two-dimensional scalar waves for simplicity)

$$\begin{aligned} E_{\text{scat}}(x, \omega) &= G_0(x, x', \omega) \alpha(\omega) E_{\text{inc}}(x', \omega) \\ &= \frac{i\omega^2}{4} H_0^{(1)}(|x - x'| \omega / c) \alpha(\omega) e^{i\omega(\hat{k} \cdot x') / c}, \end{aligned}$$

where G_0 is the free-space Green's function and $H_0^{(1)}$ is the Hankel function of the first kind. The real and imaginary parts of this function *also* are not Hilbert transforms of each other. Why is this true if scattering is a causal process? It is because a Hilbert transform requires other conditions like no poles on the real line and sufficient asymptotic decay. For a scatterer with $\alpha(\omega) = \chi(\omega) \sim -\frac{1}{\omega^2}$, the high-frequency scaling of the scattered field in this simple example is (taking a point x such that $x - x'$ is parallel to \hat{k})

$$E_{\text{scat}}(\omega) \sim \frac{1}{\sqrt{\omega}} e^{i\omega \hat{k} \cdot (x - x') / c}$$

This does not decay sufficiently fast. Moreover, suppose one multiplied this field by the prefactor like $1/\omega^2$ that preserves the correct symmetry about the imaginary-frequency axis and ensures sufficient decay? Then this would create a non-simple pole at zero frequency, significantly altering the relationship between the real and imaginary parts. Scattered fields from scatterers with nonzero size and non-symmetric shape, like that in Fig. 1(b), have even more complex asymptotic response. Scattered fields generally are not oscillators, and, indeed, looking at the traces in Fig. 1(b), one cannot discern any “oscillator-like structure.”

By contrast, the first three traces in Fig. 1(c) are “true” oscillators, in the sense that they are represented by linear combinations of Drude-Lorentz oscillators with positive, real-valued coefficients. The reviewer agrees.

What about the second set of three traces in Fig. 1(c), corresponding to the off-diagonal terms in the T matrix? Those are also linear combinations of Drude-Lorentz oscillators, but now with complex-valued coefficients. The consequence of the complex values is that the real and imaginary parts “switch roles.” But there is still clearly visible structure in those traces in Fig. 1(c): the peaks of the real and imaginary parts tend to coincide (as expected for any phase-shifted single Drude-Lorentz oscillator), one sees “transitions” in one of the parts correlating with Lorentzian-like bumps in its counterpart, etc. Hence we use the term “oscillator-like” to describe these traces.

(A phase-shifted oscillator is not *sufficient* for our full description of the T matrix as *matrix-valued oscillators*, which goes one step further: the complex-valued coefficients of the off-diagonal parts are constrained so that every diagonal block of the matrix coefficient is positive

semidefinite, but this is not detectable in the traces. Hence why we also plot the eigenvalues in Fig. 1(d.)

Our use of the term “oscillator-like” in the caption of Fig. 1 is a qualitative description that we feel is appropriate. For those who prefer more rigorous quantitative definitions, the representation theorem of Eq. 6 is our ultimate definition of oscillator-like. Scattered fields such as those in Fig. 1(b) simply do not have such a representation.

We have added a discussion of this idea in the main text (everything after the first sentence is new):

The lineshapes of the T-matrix elements closely mimic the Drude–Lorentz- like behavior of electronic transitions, but they arise not from real material oscillators, but from complex wave-scattering behavior itself. The first three traces of Fig. 1(c) clearly show positive imaginary parts of varying widths, and real parts that transition from minima to maxima between the peaks of the imaginary parts, then transitioning back to minima where the imaginary parts peak. Hence the peaks tend to coincide (with the real-part peak slightly preceding the imaginary-part peak), and the characteristic lineshapes might be described as minima-to-maxima-to-transition for the real parts and Lorentzian-like for the imaginary parts. The second set of three traces in Fig. 1(c) don't have exactly this pattern, because they have complex-valued coefficients that mix the real and imaginary parts. But their underlying “oscillator-like” structure is still visible: one still sees peaks in one part nearly coinciding with (but slightly preceding) peaks in the counterpart, as well as Lorentzian-like lineshapes in one part being paired with minima-to-maxima-to-transition lineshapes in the other. By contrast, no such structure arises in the scattered fields of Fig. 1(b), because they simply do not have a representation resembling Eq. 6.

5. The supplemental material Sec. VI states that five sample locations are selected, but only three are listed and used.

Thank you again for the careful reading. Indeed, there are only three selected locations. We corrected the typo “5” into “3” in the SM.

Reviewer #2:

If I may point at one minor shortcoming in presentation that I consider it would be a good idea to fix, then it is that for Figure 2c and d it is nowhere specified what the "optimal 2D heterostructure" and "optimal Drude" structures entail. A specification in the supplement of household details (e.g., for the T=300 K case, what are these quoted optimal structures, perhaps how they were found/how the calculation was done) would solve this. The caption should explain also what the asterisks in Fig. 2c specify.

We thank the reviewer for this thoughtful suggestion. We changed the last sentence of the caption of Figure 2 to the following sentences:

(c) Spectral HTC of state-of-the-art materials corresponding to the rightmost two materials in panel (b), the optimal bulk Drude material (red) and optimal 2D heterostructure (blue). Asterisks mark frequencies where spectral HTC peak, which coincide with our T matrix bound prediction (grey). (d) Peak-HTC frequency of optimal material (red and blue asterisks) and of bound prediction (grey) for a range of temperatures. The state-of-the-art performances can be achieved precisely at the optimal frequencies identified by our approach.

We also add section VII and the following details to the supplementary material:

Section VII. Optimal frequencies of NFRHT for state-of-the-art materials

In Ref. 49 we study optimal bulk Drude materials, deriving a “near-field Wien’s law” and identifying peak spectral-HTC frequencies for such materials. The peak-HTC frequencies for the optimal bulk material (red asterisks in Fig. 2(d)) are $\omega_{\text{opt}} = 2.57 \frac{k_B T}{\hbar}$ according to the near-field Wien’s law, where \hbar is reduced Planck’s constant and k_B is Boltzmann constant.

Next, we studied optimal 2D heterostructures. We optimize over 2D materials with different in-plane conductivities, each parametrized by a combination of resonance frequencies and material loss rates. Furthermore, multiple different layers of 2D materials directly stacked together constitute 2D heterostructures and we focus on optimizing those with 1,2 and 3 different monolayers. We find optimal NFRHT efficiency is achieved with a single optimal layer of 2D material, and multiple stackings do not perform better. The spectral response of this structure as well as that of the optimal bulk Drude material are shown in Fig. 2(c). The exact data for optimal frequencies (red and blue asterisks in Fig. 2(d)) are listed below:

Temperature (K)	Optimal bulk Drude (eV)	Optimal 2D heterostructure (eV)
100	0.0222	0.0227
200	0.0444	0.0455
300	0.0665	0.0683
400	0.0887	0.0912
500	0.1109	0.1141
600	0.1331	0.1370
700	0.1552	0.1604
800	0.1774	0.1838
900	0.1996	0.2072
1000	0.2218	0.2291
1100	0.2440	0.2511
1200	0.2661	0.2730

REVIEWERS' COMMENTS

Reviewer #1 (Remarks to the Author):

The authors have addressed all of my concerns.